# Allotrope-dependent activity-stability relationships of molybdenum sulfide hydrogen evolution electrocatalysts

Daniel Escalera-López [1] ✉, Christian Iffelsberger[2], Matej Zlatar [1,3], Katarina Novčić[2], Nik Maselj[4,5], Chuyen Van Pham[1], Primož Jovanovič[4,5], Nejc Hodnik [4,5], Simon Thiele [1,3], Martin Pumera[2,6,7,8] & Serhiy Cherevko [1] ✉

Molybdenum disulfide ($MoS_2$) is widely regarded as a competitive hydrogen evolution reaction (HER) catalyst to replace platinum in proton exchange membrane water electrolysers (PEMWEs). Despite the extensive knowledge of its HER activity, stability insights under HER operation are scarce. This is paramount to ensure long-term operation of Pt-free PEMWEs, and gain full understanding on the electrocatalytically-induced processes responsible for HER active site generation. The latter are highly dependent on the $MoS_2$ allotropic phase, and still under debate. We rigorously assess these by simultaneously monitoring Mo and S dissolution products using a dedicated scanning flow cell coupled with downstream analytics (ICP-MS), besides an electrochemical mass spectrometry setup for volatile species analysis. We observe that $MoS_2$ stability is allotrope-dependent: lamellar-like $MoS_2$ is highly unstable under open circuit conditions, whereas cluster-like amorphous $MoS_{3-x}$ instability is induced by a severe S loss during the HER and undercoordinated Mo site generation. Guidelines to operate non-noble PEMWEs are therefore provided based on the stability number metrics, and an HER mechanism which accounts for Mo and S dissolution pathways is proposed.

The hydrogen economy, initially envisioned by Bockris in the late seventies[1], is now coming to shape as a viable means to decarbonise the energy sector. Besides the ever-growing scarcity of crude oil and environmental issues with fossil fuel utilization, pressing geopolitical conflicts have motivated policymakers and engaged stakeholders to implement initiatives such as DoE's 'Hydrogen Shot', whereby widespread use of renewable hydrogen production technologies such as proton exchange membrane water electrolysers (PEMWEs) are planned. Albeit a reliable technology, large-scale deployment of PEMWEs is currently bottlenecked by the scarcity and high price of the state-of-the-art platinum group metals (PGMs) used to catalyse both oxygen evolution and hydrogen evolution reactions (HER)[2–4]. To alleviate this issue, novel HER catalysts based on earth-abundant, non-noble metals have been explored to drastically reduce PGM contents in PEMWEs[5,6]. Among these, molybdenum disulfide ($MoS_2$) presents HER performances closely matching those of Pt and has already been

[1]Helmholtz-Institute Erlangen-Nürnberg for Renewable Energy (IEK-11), Forschungszentrum Jülich GmbH, Cauerstrasse 1, 91058 Erlangen, Germany. [2]Future Energy and Innovation Technology, Central European Institute of Technology, Brno University of Technology, Purkiňova 656/123, 61200 Brno, Czech Republic. [3]Department of Chemical and Biological Engineering, Friedrich-Alexander-Universität Erlangen-Nürnberg, Egerlandstrasse 3, 91058 Erlangen, Germany. [4]Department of Materials Chemistry, National Institute of Chemistry, Hajdrihova 19, 1000 Ljubljana, Slovenia. [5]Faculty of Chemistry and Chemical Technology, University of Ljubljana, Večna pot 113, 1000 Ljubljana, Slovenia. [6]Energy Research Institute @ NTU (ERI@N), Research Techno Plaza, X-Frontier Block, Level 5, 50 Nanyang Drive, Singapore, Singapore. [7]Department of Medical Research, China Medical University Hospital, China Medical University, No. 91 Hsueh-Shih Road, Taichung 40402, Taiwan. [8]Faculty of Electrical Engineering and Computer Science, VSB - Technical University of Ostrava, 17. listopadu 2172/15, 70800 Ostrava, Czech Republic. ✉e-mail: d.escalera@fz-juelich.de; s.cherevko@fz-juelich.de

implemented in PEMWE cathodes[7–9]. While a plethora of works reported detailed insights on the influence of crystalline phase (i.e., 1 T, 2H and 3 R, presenting different Mo-S atomic arrangements and stacking across a long-range periodical structure)[10–12], degree of crystallinity/disorder[13–15] and edge-to-basal plane ratio[16–19] in the HER electrocatalytic activity, the intrinsic stability of $MoS_2$ under HER operating conditions is rarely (if ever) assessed besides indirect electrochemical metrics or identical location transmission electron microscopy[20].

First evaluated on crystalline $MoS_2$[21] and more recently on $[Mo_3S_{13}]^{2-}$ cluster-based amorphous molybdenum sulfide $MoS_{3-x}$[7] allotrope (presenting distinct Mo-S atomic arrangement but no long-range periodical order found in crystalline structures), a stark stability difference was observed between hydrogen-evolving and non-HER conditions (open circuit potential, close to 0 $V_{RHE}$), later ascribed to a large thermodynamic decomposition driving force[22]. Although insightful, these reports do not provide a full picture of the overall material stability, as only Mo dissolution rates were monitored. This is timely from the HER mechanism standpoint, as sulphur vacancy formation in crystalline $MoS_2$[23–28], as well as sulphur loss in amorphous $MoS_{3-x}$ via a structural transformation to $MoS_2$[29–34], are directly responsible for the generation of HER active sites. The specific HER active site nature is, however, still under debate. Initially acknowledged to be S in crystalline $MoS_2$[35] and $MoS_{3-x}$[36], recent reports have postulated an undercoordinated Mo hydride as the universal site across all Mo-based electrocatalysts[37–40]. To fully uncover these discrepancies, and clarify the dissolution-assisted pathways responsible for $MoS_x$ electrocatalytic activity, simultaneous monitoring of both Mo and S species is required under conditions relevant for PEMWEs. Indeed, if operated in conjunction with intermittent renewable power supply, unforeseen oxidative potentials can be exerted at PEMWE cathodes yielding $MoS_2$ corrosion[41–44].

To fill this knowledge gap, we assessed the activity and stability across representative $MoS_2$ materials by simultaneously monitoring Mo and S dissolution with a scanning flow cell coupled to an inductively-coupled plasma spectrometer (SFC-ICP-MS), as well as the volatile products produced during HER by means of electrochemical mass spectrometry (EC-MS). Our study will predominantly focus on electrodeposited amorphous molybdenum sulfide thin films, anodically and cathodically-electrodeposited molybdenum sulfide (a-$MoS_{3-x}$ and c-$MoS_2$, respectively), as both allotrope and S-to-Mo stoichiometry can be easily tuned by the deposition parameters[45,46]. In addition, two nanopowdered $[Mo_3S_{13}]$-based cluster catalysts will be studied, due to

their previous implementation in non-noble PEMWE cathodes[8,47]: the highly-active $[Mo_3S_{13}]^{2-}$ self-standing clusters, and $[Mo_3S_{13}]^{2-}$ anchored to N-doped carbon nanotubes (MoS$_x$-N-CNT).

These materials will allow to draw parallels in activity-stability metrics across $[Mo_3S_{13}]^{2-}$-containing catalysts, but also evaluate the impact of $MoS_2$ allotrope in the stability window, as well as the HER-induced activation and degradation mechanisms. Our results corroborate that the lamellar-like c-$MoS_2$ is stable under HER potentials, but its durability is severely hampered under intermittent operation by Mo moieties: under non-HER potentials, these are intrinsically unstable. In addition, activation of the $[Mo_3S_{13}]^{2-}$ cluster-based catalysts under HER operating potentials is due to selective S loss. This eventually triggers Mo dissolution by formation of undercoordinated Mo sites as suggested by post-mortem X-ray photoelectron spectroscopy measurements (XPS), which in turn compromises the overall stability. Overall, a-$MoS_{3-x}$ presents the best trade-off between activity and stability if noble metal-free cathode PEMWEs were to operate under constant load, where anodic potentials are avoided. We envision that the results presented will provide clear guidelines for the operation of $MoS_2$–based PEMWEs, transferrable to other promising PGM-free acidic HER electrocatalysts.

## Results and Discussion
### Study of cathodic stability window

For ease of interpretation, we firstly describe the structure of the MoS$_x$ catalysts employed in this work (see Fig. 1 for visual representation of $MoS_2$ materials, and Section S1 for physicochemical characterization). Cathodically-electrodeposited films (c-$MoS_2$) present a lamellar-like structure analogous to crystalline $MoS_2$, whereas anodically-electrodeposited films (a-$MoS_{3-x}$) present a coordination polymer structure conformed by $[Mo_3S_{13}]^{2-}$ units yielding a S-to-Mo stoichiometry of $MoS_{3-x}$ ($x$ being ~ 0.2 for our pristine film, see Table S2). The cluster-based structure of a-$MoS_{3-x}$, besides the different stoichiometry and morphology, presents four different S chemical environments:

- unsaturated $S^{2-}$ ($S^{2-}_{unsat}$), which chemically binds adjacent $[Mo_3S_{13}]^{2-}$ clusters;
- terminal $S_2^{2-}$ ($S_2^{2-}_{term}$), directly bound to a single Mo unit per cluster;
- bridging $S_2^{2-}$ units ($S_2^{2-}_{bridg}$), bound to two Mo units per cluster;
- apical $S^{2-}$ ($S^{2-}_{ap}$), at the centre of the cluster bound to three Mo units.

**a)** Cathodically-electrodeposited molybdenum sulfide (c-$MoS_2$)

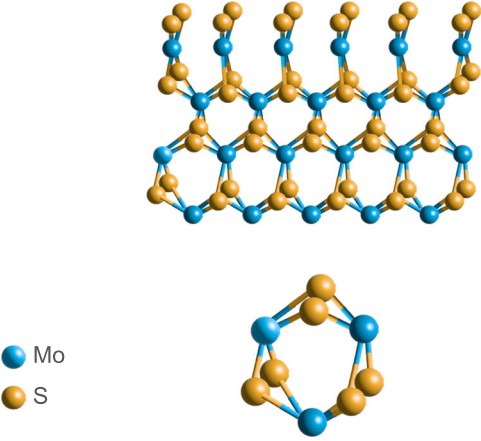

● Mo
● S

**b)** Anodically-electrodeposited molybdenum sulfide (a-$MoS_{3-x}$)

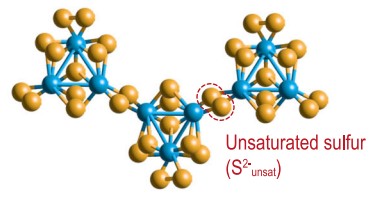

Unsaturated sulfur ($S^{2-}_{unsat}$)

$[Mo_3S_{13}]^{2-}$ cluster

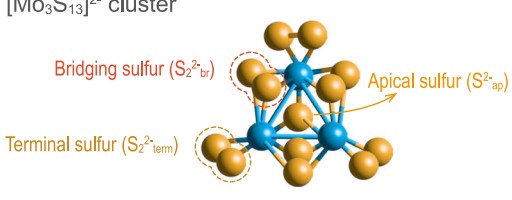

Bridging sulfur ($S_2^{2-}_{br}$)
Apical sulfur ($S^{2-}_{ap}$)
Terminal sulfur ($S_2^{2-}_{term}$)

**Fig. 1 | Chemical structure of studied MoS$_x$ allotropes.** Schematic representation of the structure for the two types of MoS$_x$ catalysts employed, based on previous reports: (**a**) c-$MoS_2$ and (**b**) a-$MoS_{3-x}$. Below, the two different building block units are visualized. The different S chemical environments found in a-$MoS_{3-x}$ films and self-standing $[Mo_3S_{13}]^{2-}$ clusters are highlighted by dashed ellipses and/or arrows. Mo and S atoms are shown in blue and yellow, respectively.

On the other hand, the self-standing $[Mo_3S_{13}]^{2-}$ clusters present 3 $S_2^{2-}{}_{term}$, 3 $S_2^{2-}{}_{bridg}$ and one $S^{2-}{}_{ap}$ per cluster, highly resembling the a-$MoS_{3-x}$ films.

Anodically-electrodeposited molybdenum sulfide films (a-$MoS_{3-x}$) are widely known to outperform their cathodic film counterparts (c-$MoS_2$) towards the HER after electrochemical activation, an activity break-in which typically consists of cycling under HER potentials[7,48]. During activation, a-$MoS_{3-x}$ undergoes a structural transformation generally ascribed to the loss of sulphur sites[29–33,49]. This is showcased in Fig. 2, where linear sweep voltammograms were recorded before and after electrochemical cycling: a 7-fold geometric current increase for a-$MoS_{3-x}$ was observed at −0.25 $V_{RHE}$, whereas c-$MoS_2$ activity did not change. After electrochemical conditioning, a-$MoS_{3-x}$ presented a ca. 100 mV lower onset potential with improved HER kinetics (Tafel slope $b \approx 65 \pm 5$ mV dec$^{-1}$ vs. $75 \pm 5$ mV dec$^{-1}$ in c-$MoS_2$), in agreement with previous reports[46]. If only structure-activity relationships were to be considered, as is the case of the majority of electrocatalytic studies, a-$MoS_{3-x}$ would be the preferred $MoS_x$ catalyst. However, the origin of the a-$MoS_{3-x}$ active sites and their stability under HER potentials would be therefore neglected, paramount for HER mechanistic understanding and long-term operational viability.

To uncover further details on the electrochemical activation process and their associated dissolution processes, successive cyclic voltammograms (CVs) were recorded towards cathodic potentials with increasing lower potential limit (LPL) values. By performing these experiments in our SFC-ICP-MS setup, both Mo and S dissolution were simultaneously monitored to track their potential-dependent stability. For a-$MoS_{3-x}$ and c-$MoS_2$ electrodeposited thin films, CVs at cathodic potentials were recorded from 0 $V_{RHE}$ to LPLs in the range −0.1 to −0.3 V, with successive LPL increments of −50 mV. Figure 3 compiles the resulting potential profiles (Fig. 3a) and the associated time-dependent Mo (Fig. 3b) and S (Fig. 3c) dissolution profiles detected downstream at the ICP-MS for c-$MoS_2$ and a-$MoS_{3-x}$.

In agreement with our previous investigations of $MoS_x$-based catalysts[7,21], both materials present a prominent dissolution peak once the electrolyte touches the thin films at 0 $V_{RHE}$. Namely, Mo dissolution upon electrolyte contact is the highest detected and in the range of 50–100 ng cm$^{-2}$ (ca. 0.1% of the initial Mo loading per catalyst), regardless of the following electrochemical protocol employed. In contrast, contact S dissolution is two-three orders of magnitude lower: ca. 0.2 ng cm$^{-2}$. Preferential Mo contact dissolution can be ascribed here to the high-valence Mo surface states ($Mo^{5+}$ as found in $Mo^{5+}O_xS_y$, as well as $Mo^{6+}$ from $MoO_3$), known to be unstable once exposed to acidic electrolyte environments[48]. The slightly higher contact peak dissolution found for c-$MoS_2$ is in line with the highest content of $Mo^{5+/6+}$ surface oxides, identified in XPS measurements (see Figs. S1 and S25).

To evaluate the influence of LPL in the relative Mo/S dissolution, the ICP-MS dissolution profiles were integrated and plotted against the employed LPL values in the CVs (Fig. 4, for loading-normalized dissolution see Fig. S2), providing two main sets of conclusions. First, c-$MoS_2$ undergoes preferential Mo dissolution whilst a-$MoS_{3-x}$ preferentially loses S under HER potentials. For both materials, the preferential loss is 1–2 orders of magnitude higher with respect to the other element, which demonstrates clear differences in the cathodic dissolution pathways involved. Second, c-$MoS_2$ stabilized upon sequential cycling in HER potentials as Mo/S dissolution reached a plateau, whereas a-$MoS_{3-x}$ dissolution scaled with increasing LPL values.

The intrinsic stability of Mo and S under HER potentials, along with the effect of electrochemical pre-history, were assessed by plotting the onset potentials of dissolution of Mo and S vs. the LPLs

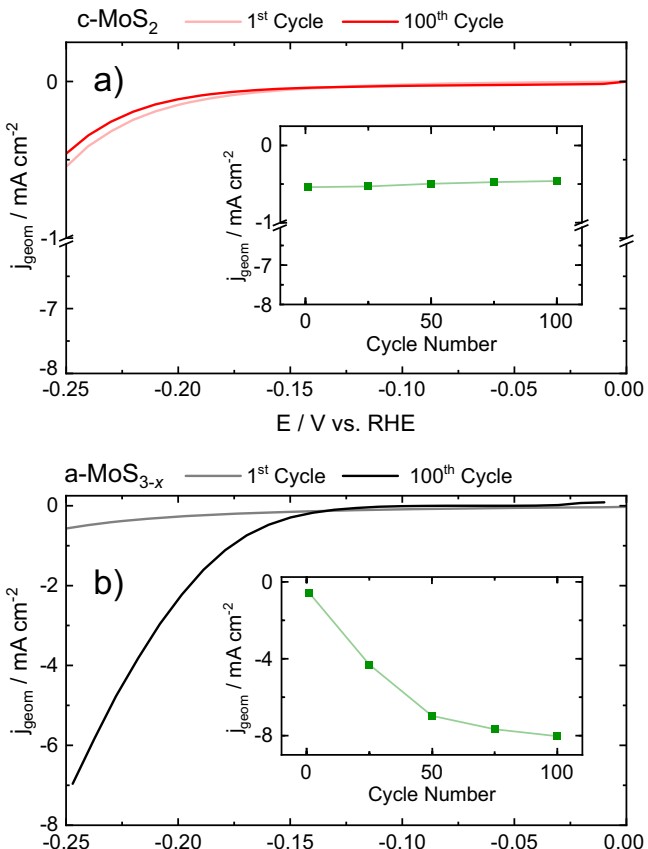

**Fig. 2 | Electrochemical activity of MoS_x allotropes.** Representative linear sweep voltammograms for (**a**) c-$MoS_2$ (red) and (**b**) a-$MoS_{3-x}$ (black) recorded before (pale) and after (dark) 100 CVs from 0 to −0.25 V vs. RHE. Inset: maximum HER current densities per cycle number. Scan rate: 5 mV s$^{-1}$.

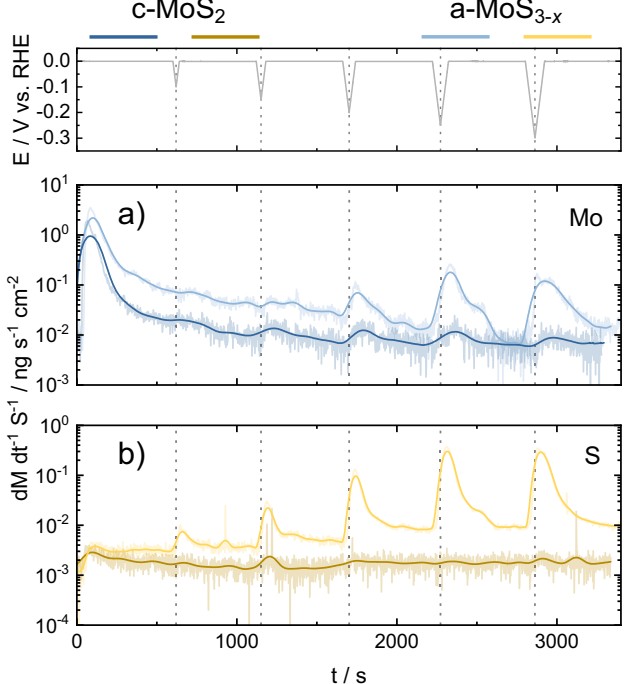

**Fig. 3 | Online ICP-MS at varying hydrogen evolution LPLs.** Online ICP-MS dissolution data obtained for successive cyclic voltammograms (CVs) with varying lower potential limit (LPL) values for (**a**) Mo (blue) and (**b**) S (yellow) in c-$MoS_2$ (dark) and a-$MoS_{3-x}$ (pale). Dissolution rates expressed as material loss (M: either Mo or S) per time and catalyst geometric area. CVs recorded from 0 $V_{RHE}$ to UPLs in the range $-0.1 \leq E_{UPL} \leq -0.3$ V. Scan rate: 5 mV s$^{-1}$.

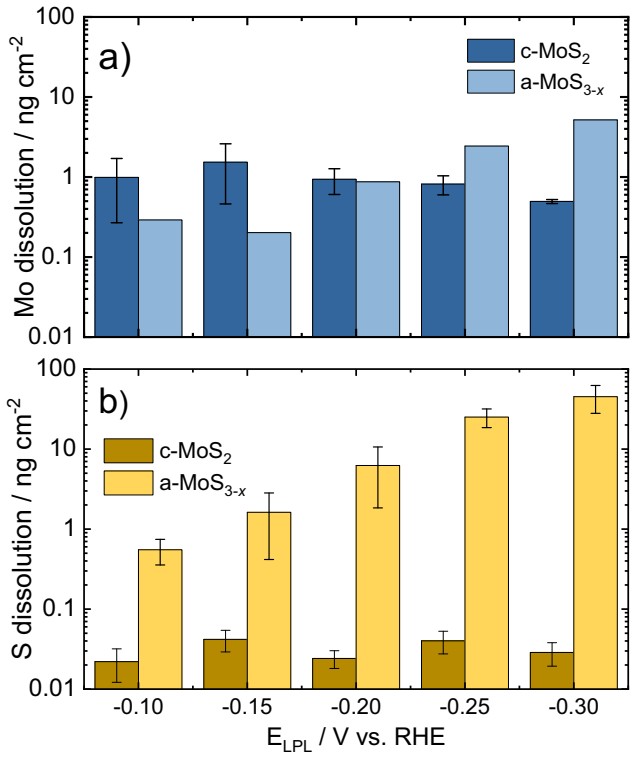

**Fig. 4 | Total integrated Mo and S dissolution at varying LPLs.** Graphical representation of total integrated dissolution of (**a**) Mo (blue) and (**b**) S (yellow) as a function of the LPL. For electrochemical protocol, see Fig. 1. Scan rate: 5 mV s⁻¹.

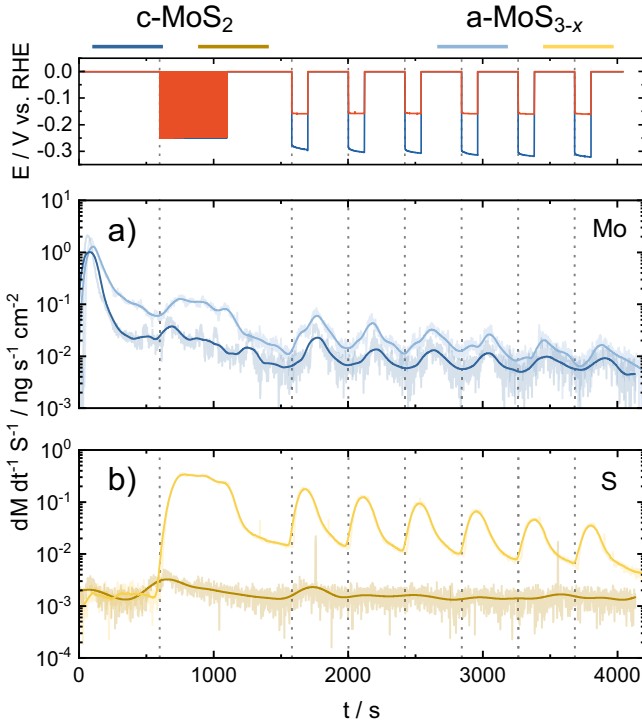

**Fig. 5 | Online ICP-MS data during HER start-up/shutdown.** Online ICP-MS dissolution data obtained for (**a**) Mo (blue) and (**b**) S (yellow) in c-MoS₂ (dark) and a-MoS₃₋ₓ (pale) during start-up/shut-down stress tests. E vs. t profile shown in blue and red for c-MoS₂ and a-MoS₃₋ₓ, respectively. Dissolution rates expressed as material loss (M: either Mo or S) per time and catalyst geometric area. Pre-conditioning: 100 CVs, 0 to −0.25 V$_{RHE}$, 100 mV s⁻¹.

(Fig. S3). For both catalysts, the obtained Mo dissolution onsets are almost identical regardless of the MoS₂ allotrope studied. In addition, Mo dissolution onsets shifted to more negative values during sequential cycling. This would indicate that the Mo centers are gradually stabilized under cathodic potentials and that the resulting stabilized Mo moieties should be comparable. In contrast, S dissolution onsets are clearly allotrope-dependent: for c-MoS₂ they negatively shifted with sequential cycling and had similar values to those of Mo, whereas for a-MoS₃₋ₓ they peaked at LPL of −0.15 V$_{RHE}$ and positively shifted in subsequent cycles, in excellent agreement with the volatile H₂S detection onset from identical EC-MS experiments (Fig. S3).

These findings suggest that Mo and S dissolution processes are only concomitant for c-MoS₂, and that for a-MoS₃₋ₓ an HER-induced S loss pathway is facilitated at a given overpotential. Indeed, this would be in excellent agreement with the HER pre-peak consistently reported for pristine electrodeposited a-MoS₃₋ₓ films. The pre-peak feature is only observed in the first cathodic cycle, related to an irreversible allotrope conversion via S loss to a structure similar to c-MoS₂ which is responsible for the higher HER activity[29,33,48]. Hence, our results present a direct link between a previously reported electrochemical MoSₓ allotrope conversion with a shift in the intrinsic S stability in a-MoS₃₋ₓ.

Analogous experiments performed on [Mo₃S₁₃]-based powder catalysts lead to similar conclusions to those found for a-MoS₃₋ₓ, where MoSₓ-N-CNT presented the lower Mo and S dissolution onsets and highest dissolution. These are ascribed to the higher HER activity and [Mo₃S₁₃] catalyst utilization provided by the highly-conducting, percolated N-CNT support (Figs. S4, S5). This is in good agreement with previous findings of Chung et al. which correlated, for a given MoSₓ catalyst, a lower utilization (i.e. poor catalyst percolation) with worsened HER metrics due to kinetic limitations[50]. Thus, analogous to other materials, there is a correlation between faster reaction rates and dissolution kinetics[51–53].

## Study of stability under intermittent operation: start-up/shut-down

After establishing the stability windows for c-MoS₂ and a-MoS₃₋ₓ and evaluating the relative stability trends for Mo and S, we proceeded to evaluate the electrochemical stability under "start-up/shutdown" HER holds, inspired by previous ICP-MS dissolution studies on Ir-based OER[52] and non-noble metal HER catalysts[21]. In particular, we employed six consecutive start-stop cycles from −1 mA cm⁻²$_{geom}$ to 0 V$_{RHE}$, which aim to mirror the fluctuating renewable energy input under which PEMWEs would operate to produce green hydrogen[54]. The application of relatively low current densities is dictated by limitations of the on-line ICP-MS setup. These holds are preceded by a preconditioning step, aimed at activating [Mo₃S₁₃]-based electrocatalysts as was shown in Fig. 2 and previously reported[29].

Figure 5 compiles the time-dependent potential (top panel) plot along with the downstream recorded ICP-MS Mo and S dissolution profiles for c-MoS₂ and a-MoS₃₋ₓ. Side-by-side comparison of the ICP-MS signal for Mo and S during the start-up/shutdown HER cycles shows contrasting dissolution profiles. For Mo, dissolution is detected not only during the preconditioning and galvanostatic holds but also on the following 0 V$_{RHE}$ potentiostatic holds, to a lesser extent in c-MoS₂. Conversely, S dissolution is solely observed under HER operating currents.

Integration of the total and loading-normalized ICP-MS dissolution profiles (Figs. S6, S7) shows a gradual catalyst stabilization between the first and sixth HER holds. After sequential on-off cycles, Mo dissolution in c-MoS₂ is ca. 10-fold higher than S, whereas S dissolution in a-MoS₃₋ₓ is 100-fold higher than that of Mo, in line with Fig. 3. Interestingly, Mo dissolution during the 0 V$_{RHE}$ holds is more prominent than during HER potentials, and fairly constant irrespective of the number of HER holds and MoSₓ allotrope (Figs. S6–S8).

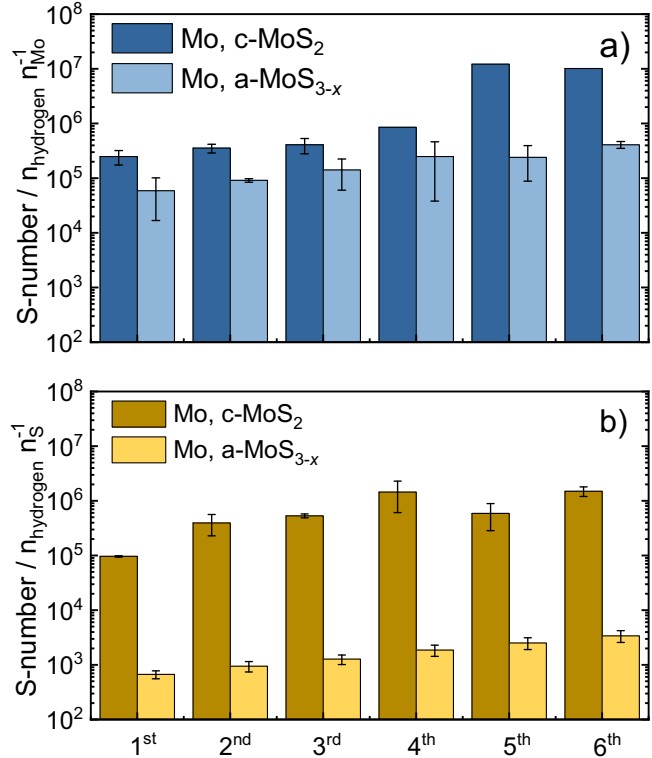

**Fig. 6 | S-numbers for Mo and S during start-up/shutdown.** Compilation of S-numbers obtained during start-up/shut-down stress tests for a) Mo (blue) and b) S (yellow) under HER potentials.

A similar "start-up/shutdown" protocol was employed for [$Mo_3S_{13}$]-based catalysts (Fig. S9), leading to comparable Mo/S dissolution trends. In line with the LPL study, the highly-conducting percolated network in $MoS_x$-N-CNT yields higher HER activity than [$Mo_3S_{13}$]$^{2-}$ after the intermittent operation. This is accompanied by higher Mo/S dissolution rates, mainly related to the 10 and 2-fold higher HER currents recorded after preconditioning for $MoS_x$-N-CNT (ca. −2 mA cm$^{-2}$) compared to [$Mo_3S_{13}$]$^{2-}$ (ca. −0.1 mA cm$^{-2}$) and a-$MoS_{3-x}$ (ca. −1 mA cm$^{-2}$, see Table S1).

In order to better evaluate the stability trends, we employed our recently proposed stability benchmarking metric, the so-called stability number (S-number). Initially reported for Ir-based oxygen evolution electrocatalysts[55], the S-number has been adopted by multiple research groups to assess gas-evolving electrocatalytic processes[56–59]. In brief, the S-number normalizes the number of evolved gas products (assuming 100% Faradaic efficiency) per atom of electrocatalyst dissolved. Higher S-numbers indicate a higher electrocatalyst stability, and vice versa. Following up on its recent use in $MoS_x$-based catalysts to estimate Mo S-numbers under HER potentials[7], we extended their use here to S (Fig. 6). In agreement with the dissolution trends, Mo S-numbers under hydrogen-evolving potentials increased during intermittent operation: from ~$10^5$ to ~$10^7$ in c-$MoS_2$, and from ~$5 \times 10^4$ to ~$4 \times 10^5$ in a-$MoS_{3-x}$. Analogously, S S-numbers increased for c-$MoS_2$ (~$9 \times 10^4$ to ~$1 \times 10^6$) and a-$MoS_{3-x}$ (~$6 \times 10^2$ to ~$3 \times 10^3$). These results corroborate the almost 10 and 100-fold higher stability of Mo compared to S in c-$MoS_2$ and a-$MoS_{3-x}$, respectively. In addition, Mo and S stability in c-$MoS_2$ are two to three orders of magnitude higher than in a-$MoS_{3-x}$, showcasing the differences in intrinsic stability between the two catalyst allotropes. Regardless of the slightly different catalyst preconditioning and loading, the S-numbers obtained for [$Mo_3S_{13}$]-based catalysts were comparable to those of a-$MoS_{3-x}$ for both Mo and S (Fig. S10). Among these catalysts, and in line with the LPL study, the best-performing $MoS_x$-N-CNT catalyst presented slightly lower

S-numbers for Mo (~$1 \times 10^4$ to ~$3 \times 10^4$) than a-$MoS_{3-x}$ (Mo: ~$2 \times 10^4$ to ~$6 \times 10^4$). This points towards the classical activity-stability trade-off: high activities are obtained at the expense of lower stabilities. Surprisingly, S S-numbers for [$Mo_3S_{13}$]-based catalysts converged during sequential on/off cycles to ~$1 \times 10^4$ (Fig. S10). This might indicate that electrochemical S loss rates are dependent on the reaction rates at every catalyst, eventually leading to a steady state. Regardless, the S-numbers obtained reinforce the hypothesis that all [$Mo_3S_{13}$]-based catalysts present equivalent stability trends only affected (to a minor extent) by catalyst utilization.

At 0 $V_{RHE}$, the S-number metric cannot be employed, as such potentials should not catalyse any gas evolution although an electrochemical charge is recorded. Thus, we employ here the so-called S-number(e$^-$) metric, which was previously proposed for electrochemical processes in which the reaction product is not volatile. Briefly, it normalizes the number of transferred electrons per atom of electrocatalyst dissolved[60]. For ease of comparison with the S-numbers at HER potentials, it is assumed that Mo dissolution undergoes a two-electron transfer process. Remarkably, the S-numbers(e$^-$) calculated for Mo during the 0 $V_{RHE}$ 'off' cycles were fairly constant and independent of the $MoS_x$ allotrope (ca. $10^3$, Fig. S8) and [$Mo_3S_{13}$]-based catalyst (ca. $10^2$, see Fig. S10). This would reinforce our previous claim that, under HER potentials, an equivalent undercoordinated Mo species is produced which is drastically destabilized at non-cathodic potentials. In contrast, inspection of the galvanostatic hold-dependent HER potentials (Fig. S8) showcases a consistently improved HER activity for a-$MoS_{3-x}$, yielding 150 mV lower potentials than c-$MoS_2$ (ca. −0.16 $V_{RHE}$ vs. ca. −0.30 $V_{RHE}$) across the intermittent stability protocol. The discrepancy between the stability and activity metrics clearly demonstrates the need to monitor both parameters to unambiguously show proof of long-term electrocatalyst durability. Alternatively, backing electrode passivation[61,62] and microbubble blockage of catalytic sites[63], among other effects, have also been reported to preclude any physically relevant conclusions from electrochemical data[4,64,65].

The aforementioned online ICP-MS experiments enable to track down the dissolved Mo and S species flown downstream by the electrolyte but provide no physical information on the remaining surface species as well as any volatile products derived from the dissolution processes. These can be uncovered by techniques such as XPS and EC-MS, used here as complementary tools to gather insights. To shed light on the impact of catalyst preconditioning and start-up/shutdown HER operation in the surface composition of the [$Mo_3S_{13}$]-based catalysts, we recorded post-mortem XPS spectra after full completion of the preconditioning and start-stop testing steps for $MoS_x$-N-CNT. Given its best performance and highest catalyst utilization, changes in the surface species present are expected to be maximized. Figure 7 displays the high-resolution Mo 3$d$ and S 2$p$ spectra of $MoS_x$-N-CNT tested under potentiodynamic preconditioning, as well as the relative abundance of each surface species present. For Mo, a drastic decrease in Mo$^{4+}$ surface content is found after the preconditioning step (68 at. %) and after start-stop HER experiments (54 at. %), at the expense of increased relative content of the oxidized Mo$^{5+}$O$_x$S$_y$/Mo$^{6+}$ species. Thus, the undercoordinated Mo sites anticipated to be present at HER potentials are directly oxidized to Mo$^{5+}$O$_x$S$_y$/Mo$^{6+}$ upon voltage reversal and environment exposure. In the case of S, intermittent HER operation yields increased presence of oxidized SO$_x^{y-}$ moieties (up to 17 at. %, labelled in Fig. 7C) with a relative S$_2^{2-}$/S$^{2-}$ ratio modified towards a higher presence of S$_2^{2-}$$_{term}$/S$^{2-}$$_{unsat}$ ligands (labelled in Fig. 7A). These results are in agreement with previous HER *in operando* $MoS_x$ composition studies, where the gradual loss of terminal S$_2^{2-}$ ligands[32], along with the cleaving of bridging S$_2^{2-}$ (labelled in Fig. 7B)[36] to yield dangling S$^{2-}$ units were proven[30]. Consequently, [$Mo_3S_{13}$]-based catalysts undergo a structural transformation, involving a gradual conversion of S and Mo moieties to undercoordinated Mo$^{3+}$ and

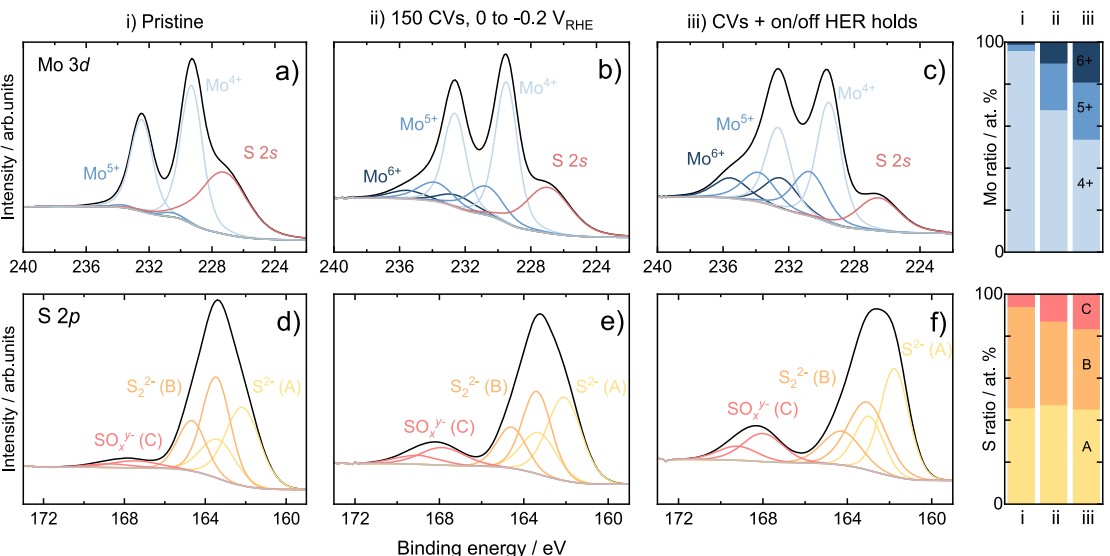

**Fig. 7 | Ex-situ XPS data on Mo₃S₁₃-NCNT before/after HER experiments.** High-resolution Mo 3*d* (**a**–**c**) and S 2*p* (**d**–**f**) XPS spectra Mo₃S₁₃-NCNT before electrochemistry (**a**, **d**) and after HER preconditioning (**b**, **e**) followed by start-up/shut-down stress tests (**c**, **f**). Labels: cumulative peak fit (black), Mo⁴⁺ 3d₅/₂:₃/₂ (light blue), MoᵃOᵦSᶜ 3d₅/₂:₃/₂ (blue), Mo⁶⁺ 3d₅/₂:₃/₂ (dark blue), S 2p₃/₂:₁/₂ (S²⁻, yellow), S 2p₃/₂:₁/₂ (S₂²⁻, orange) and S 2p₃/₂:₁/₂ (SOₓʸ⁺, red).

unsaturated $S^{2-}$, but also a significant S loss as previously reported with electrochemical quartz crystal microbalance measurements[29] and corroborated here.

Surprisingly, despite the selective S loss during activation of [Mo₃S₁₃]-based catalysts, the prominent Mo loss at 0 $V_{RHE}$ should still yield slightly S-rich S-to-Mo ratios. This is based on our back-of-the-envelope calculation from the online ICP-MS measurements, accounting for Mo/S loss versus initial catalyst loadings. As anticipated, the highly-active MoSₓ-N-CNT presents the highest degree of S enrichment (see Table S2 for values). This contrasts with the Mo-rich surface ratios found after electrochemical testing with XPS, which would suggest a compositional gradient across the catalyst.

Previous reports on a-MoS₃₋ₓ already suggested a Mo-rich surface at the activated surface, after $H_2S$ quantification by differential electrochemical mass spectrometry[34]. To clear out the discrepancy in the S-to-Mo ratio, we further monitored the volatile products derived from HER with an electrochemical mass spectrometry setup (EC-MS, Fig. 8). Indeed, $H_2S$ (m/z = 34) could be quantified for a-MoS₃₋ₓ but was not detected for c-MoS₂. We believe that the extremely low ICP-MS detection limits (sub-pg s⁻¹ cm⁻²) are beyond those of the EC-MS, preventing accurate $H_2S$ quantification evolved from c-MoS₂. Detection of gaseous $H_2S$ across the HER protocols employed here provides further proof of the a-MoS₃₋ₓ activation pathway via S loss, and the direct link between undercoordinated Mo sites and higher HER rates. This could in addition explain the S-to-Mo ratios discrepancy, as gaseous $H_2S$ could potentially diffuse through downstream tubing or even the polycarbonate body of the SFC, not yielding a 100% collection efficiency at the ICP-MS.

## Study of long-term stability: H-cell measurements

In order to validate the observed trends at application-relevant operating currents, long-term stability measurements were performed at −100 mA cm⁻² (ca. −741 mA mg$_{cat}$⁻¹) in an H-cell configuration for both freshly-prepared c-MoS₂ and a-MoS₃₋ₓ (see Figs. S23, S24 for results and S25 for physicochemical characterization). To assess the impact of intermittent operation, H-cell measurements were performed for 5 hours under constant HER load as well as under start-up/shutdown conditions (start-stop cycles from −100 mA cm⁻²$_{geom}$ to 0 $V_{RHE}$, 2 min per pulse). It was observed that only Mo dissolution could be quantitatively assessed, which leads us to conclude that under high HER

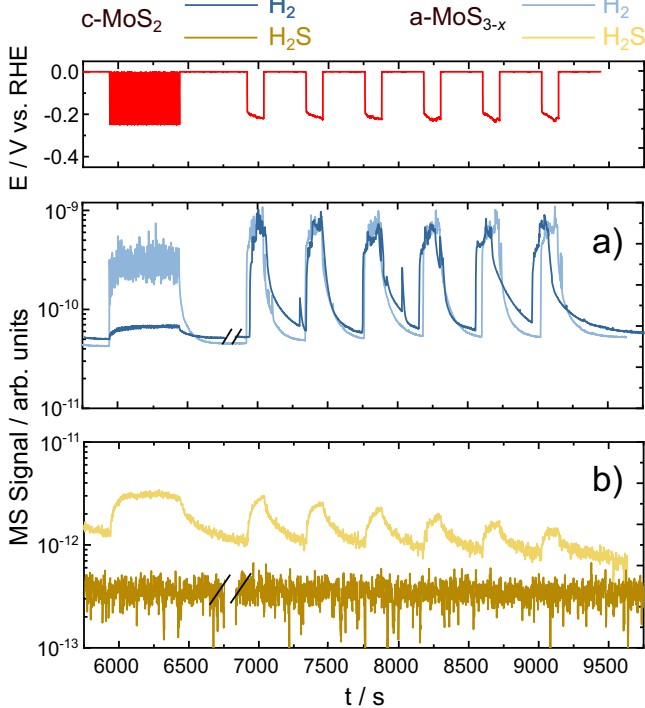

**Fig. 8 | Online volatile products detection during HER.** Electrochemical mass spectrometry (EC-MS) measurements performed on c-MoS₂ and a-MoS₃₋ₓ during start-up/shut-down stress tests. Preconditioning: 100 CVs, 0 to −0.25 $V_{RHE}$, 100 mV s⁻¹. MS signal of (**a**) gaseous $H_2$ (blue, m/z = 2) and (**b**) $H_2S$ (yellow, m/z = 34) plotted versus the stress test protocol.

reaction rates S loss predominantly takes place by gaseous $H_2S$ evolution for both catalysts, as S does not remain in the liquid phase. The resulting S-numbers obtained after Mo quantification of H-cell liquid aliquots (Fig. 9) showcase that constant operation yields 3 to 5 times higher stability for both catalysts (1.2 to 2 times higher loading-normalized dissolution, see Fig. S24 and Table S3 for values), and that the stability trends are inversed compared with those at low current

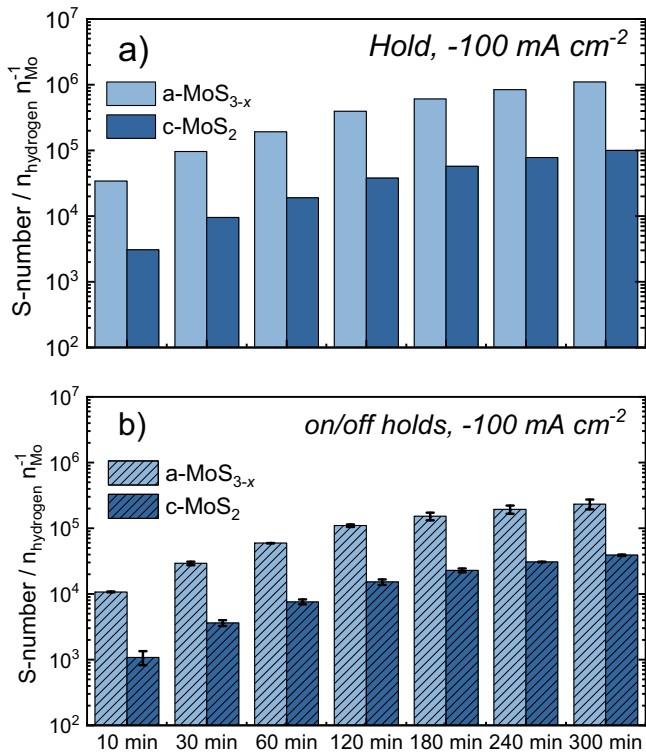

**Fig. 9 | S-numbers for long-term, high-current HER measurements.** Compilation of Mo S-numbers obtained during long-term H-cell measurements consisting of (**a**) constant HER operation (5 h, −100 mA cm$^{-2}$; left column) and (**b**) start-up/shutdown HER intermittent operation alternating (5 h, −100 mA cm$^{-2}$ to 0 V vs. RHE; right column). Labels: a-MoS$_{3-x}$ (pale blue), c-MoS$_2$ (dark blue).

densities: a-MoS$_{3-x}$ stability is almost 10-fold higher than c-MoS$_2$, regardless of the operation mode. Post-mortem, ex-situ SEM imaging demonstrated that catalyst degradation by mechanical delamination was not present in either c-MoS$_2$ (see A–C Fig. S26) or a-MoS$_{3-x}$ (see A–C Fig. S28), therefore not being responsible for the observed trends. Analogous conclusions were drawn from XRD measurements, whereby all diffractograms remained unchanged after testing (see D-F Figs. S26 and S28). In contrast, XPS and Raman measurements showcased a clear loss of the distinct $E^1_{2g}$ and $A_{1g}$ Raman modes in c-MoS$_2$ (see G-I Fig. S26) as well as an increased presence of oxidized Mo$^{5+}$O$_x$S$_y$:Mo$^{6+}$ species at the expense of Mo$^{4+}$ from MoS$_2$ (ca. 9:42 at. % after intermittent operation, see Fig. S27). Interestingly, post-mortem a-MoS$_{3-x}$ presented almost negligible Mo$^{5+}$O$_x$S$_y$:Mo$^{6+}$ surface contents (ca. 2:1 at. %, see Fig. S29) which, along with the loss of S$_2^{2-}$$_{bridg}$/S$_2^{2-}$$_{term}$/S$^{2-}$$_{ap}$ Raman modes (ca. 550, 520 and 450 cm$^{-1}$), the well-reported appearance of both a band at 430 cm$^{-1}$ (resembling $A_{1g}$) and MoO$_x$-related Raman features upon environment exposure (broad band at 800–1000 cm$^{-1}$), corroborate the structural transformation to the MoS$_{2-x}$ structure via S loss and undercoordinated Mo site generation. Therefore, the initially high c-MoS$_2$ stability observed worsens at high HER reaction rates in contrast with a-MoS$_{3-x}$, which presents similar Mo S-numbers at low and high HER rates ( ~ 10$^5$–10$^6$, see Figs. 6, 9, S19 and S21). We must note, however, that Mo S-numbers might be overestimated given the S loss via volatile H$_2$S.

In light of the stark differences in both activity and stability of MoS$_x$ catalysts according to their allotrope and operating conditions, we must turn our attention to several aspects. If our analysis was to be purely based on structure-performance relationships, it is clear that the [Mo$_3$S$_{13}$]-based catalysts (a-MoS$_{3-x}$, pristine [Mo$_3$S$_{13}$]$^{2-}$ clusters and MoS$_x$-N-CNT) present a superior HER performance compared to the crystalline-like c-MoS$_2$, MoS$_x$-N-CNT being the best performer due to

the highest MoS$_x$ catalyst utilization. However, our stability measurements present three main sets of conclusions. First, under HER potentials S dissolves preferentially for [Mo$_3$S$_{13}$]-based catalysts, and its dissolution is 100 times higher than Mo. Second, all [Mo$_3$S$_{13}$]-based catalysts present 100-fold lower stability under HER potentials than c-MoS$_2$ at low HER reaction rates, based on the S-number metric: for Mo: S ~ 10$^5$ vs. 10$^7$; for S: S ~ 10$^4$ vs. 10$^6$. At high reaction rates, however, a-MoS$_{3-x}$ stability is almost 10-fold higher than c-MoS$_2$ (S ~ 10$^6$ vs. 10$^5$ under constant HER hold). Third, preferential Mo dissolution takes place at 0 V$_{RHE}$ upon reversal from hydrogen-evolving potentials, regardless of the MoS$_x$ catalyst employed, yielding a similar S-number(e$^-$) at 0 V$_{RHE}$. This contrasts with the much higher Mo dissolution onset potentials during cyclic voltammetry experiments observed for all pristine catalysts (ca. 0.3 V$_{RHE}$, see Figs. S11–16 Section S2). Consequently, we should address the role of the surface species present under HER potentials in the stability trends (structure-stability relationships), and the overall implications of MoS$_x$ catalyst structure in the long-term HER activity and stability (structure-activity-stability relationships).

**Analysis of structure-stability relationships**

Although MoS$_2$ instability at anodic potentials was initially ascribed to a large thermodynamic driving force towards its decomposition[22], both LPL and upper potential limit (UPL) studies in this work (see Section S2 for the latter) showcase a clear electrochemical prehistory dependence: for all MoS$_x$ catalysts studied, Mo centers exhibited greater instability after exposure to HER potentials than during direct electro-oxidation treatment. This clearly correlates the role of S loss during electrochemical activation in [Mo$_3$S$_{13}$]-based catalysts, i.e. conversion of MoS$_{3-x}$ to MoS$_{2-x}$ as confirmed by XPS and EC-MS measurements, with both activity enhancement and formation of unstable undercoordinated Mo species. A similar activity enhancement was observed for MoS$_2$ after S vacancy formation[15,23,66], generally ascribed to strain effects which yield optimal hydrogen binding at undercoordinated Mo sites[25,26,67] or a 2H/1T phase transition[67]. Thus, the higher HER activity provided by these abundant undercoordinated Mo sites is at the expense of overall lower catalyst stability due to the post-HER Mo dissolution.

With regards to the HER active sites, recent experimental evidence from Bau et al. has directly confirmed the formation of a Mo$^{3+}$ hydride under HER potentials in a-MoS$_{3-x}$ using electron paramagnetic resonance in organic media, and indirectly correlated its abundance across different MoS$_2$ materials with the intensity of the reductive pre-peak responsible for S loss[39]. Such Mo-H state, previously proposed by theoretical[37] and experimental works[32,38] as the HER active site in MoS$_2$ materials, were recently suggested to also be universally responsible for all Mo-based HER electrocatalysis[40]. This evidence would be in direct opposition with substantial reports which correlate HER activities with Mo-edge site length[16], higher presence of proton-accepting S moieties[68] or improved phase-dependent hydrogen binding energies[26].

We believe that, indeed, undercoordinated Mo$^{3+}$ hydride sites are formed under HER potentials across all MoS$_x$ catalysts tested. The proposed HER mechanism (based upon references 32 and 39), along with the Mo and S dissolution pathways, are displayed in Fig. 10 (see section S3 for corresponding redox equations). For a-MoS$_{3-x}$, hydrogen could be evolved either via a Mo$^{3+}$ hydride (II-III in Fig. 10)[39] or a Mo$^{5+}$ hydride pathway (IV-VI in Fig. 10)[32]. The formation of the active hydride sites would take place via a concomitant loss of apical S$^{2-}$ and terminal S$_2^{2-}$ ligands[32], along with the cleaving of bridging S$_2^{2-}$ releasing volatile H$_2$S (I-II in Fig. 10, a-MoS$_{3-x}$) as shown in our post-mortem Raman measurements (Fig. S28) and S-to-Mo XPS ratios (Table S4). For c-MoS$_2$, undercoordinated Mo$^{3+}$ sites would be obtained after the loss of unsaturated S$^{2-}$ sites (I'-II' in Fig. 10, c-MoS$_2$). For both a-MoS$_{3-x}$ and c-MoS$_2$, cathodic dissolution of Mo is proposed to take place either via

a direct cleaving of S-Mo$^{3+}$-H moieties or via a sequential S loss yielding protonated Mo$^{3+}$-H species (IV'-VI', c-MoS$_2$; VII-VIII, a-MoS$_{3-x}$). Given the almost identical Mo/S dissolution onsets found in c-MoS$_2$, the Mo dissolution pathway can take place right after Mo$^{3+}$ hydride formation. The positive shift in S dissolution onset potential observed for a-MoS$_{3-x}$ beyond −0.15 V$_{RHE}$ in our LPL study (Fig. S3), as well as preferential S dissolution, would indicate that specific reaction pathways facilitate S loss over Mo, accentuated at higher HER reaction rates as Mo greatly stabilized and S was preferentially lost at the a-MoS$_{3-x}$ near-surface (see Table S4). We hypothesize that these can be due to 1) constant formation of Mo$^{3+}$ hydride sites and HER electrocatalysis with low Mo loss (I-III), 2) preferential HER electrocatalysis via the Mo$^{5+}$ hydride pathway (V-VI) or 3) bond weakening of neighbouring S ligands after the formation of Mo hydride sites favouring their cleaving and loss. Regardless of the catalyst and HER rates, the Mo hydride sites are inherently unstable at non-HER potentials and are oxidized upon voltage reversal to soluble Mo species in a complex electro-oxidation process[22,48], which are responsible for the similar S-number(e$^-$) values at 0 V$_{RHE}$ after operating under equivalent HER rates (i.e. comparable mass-normalized current densities).

Based on the Mo-H activity universality hypothesis, HER activity in Mo-based electrocatalysts is dictated by the amount of Mo hydride sites, if these have the same chemical environment. The Mo$^{4+}$/Mo$^{3+}$ reduction onset potential was indeed attributed to being the universal HER activity descriptor for Mo-based catalysts[40], which is in line with the almost identical cathodic Mo dissolution onset potentials observed for all MoS$_x$ catalysts (Figs. S3, S4). This would initially indicate that the active sites have the same chemical environment. At equivalent chemical environments, the stability of all MoS$_x$ catalysts should be comparable under identical HER rates (imposed in our start-up/shut-down stress tests), as the intrinsic stability of the Mo-H species would dictate the Mo stability trends. In other words, all Mo-based catalysts should have the same Mo S-numbers under identical mass-normalized HER current densities regardless of the amount of active sites if Mo/S dissolution is solely caused by HER. However, the different Mo S-numbers obtained for c-MoS$_2$ compared to a-MoS$_{3-x}$ at low and high HER rates (100-fold higher and 10-fold lower, see Figs. 6 and 9) would contradict this hypothesis: either Mo$^{3+}$ hydride sites would be (de)stabilized (i.e. different intrinsic activity) or an alternative HER pathway would be involved.

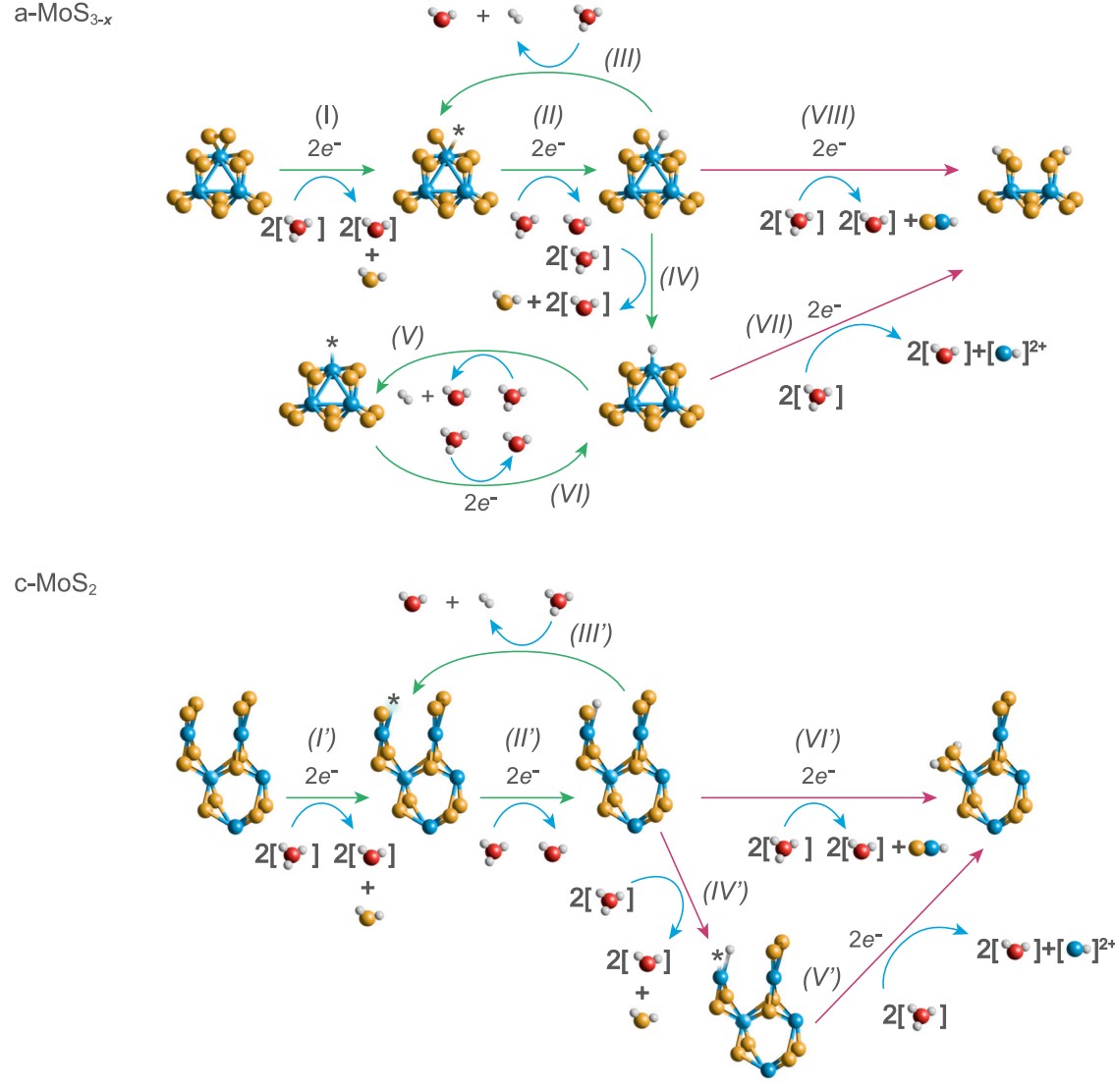

**Fig. 10 | Proposed HER and dissolution pathways.** Graphical representation of the hydrogen evolution mechanism for a-MoS$_{3-x}$ (top) and c-MoS$_2$ (bottom) driven by the Mo$^{3+}$ hydride formation as proposed by Tran et al. and Bau et al. (refs. 32,39., green arrows), along with the proposed cathodic dissolution pathways for Mo and S (red arrows). For clarity purposes the electrochemical loss of apical sulphur in a-MoS$_{3-x}$ is omitted (1-proton, 1-electron transfer process), which takes place in either a previous step or coupled to terminal sulphur loss (2-proton, 2-electron transfer process). Mo, S, O and H atoms are shown in blue, yellow, red and white, respectively.

The intrinsic $MoS_x$ active site activity should be therefore discussed. In order to have the same amount of Mo-H as a-$MoS_{3-x}$, c-$MoS_2$ should dissolve the same amount of S at equivalent HER rates. This is not the case, given the 100-fold lower S dissolution in c-$MoS_2$ under HER potentials at low HER rates (Fig. 6). At equivalent HER rates, higher turnover frequencies per available Mo-H site would then be required. Indeed, a previous study on temperature-dependent $MoS_x$ activity reported 10-fold higher turnover frequencies (TOF) for polycrystalline $MoS_2$ than amorphous $MoS_{3-x}$ despite of higher Tafel slopes[14]. While having a higher number of active sites, the poorly-conducting yet more active a-$MoS_{3-x}$ presents lower TOFs. In this work we observed that Mo centers in a-$MoS_{3-x}$ are 100-fold more unstable compared with c-$MoS_2$ at low HER rates (Fig. 6), while retaining a higher HER activity (Figure S8). This would require, at least, a 1000-fold higher density of Mo-H active sites in a-$MoS_{3-x}$ to compensate for the performance losses (i.e. Mo loss) if Mo sites were only responsible for the HER activity. At high HER rates, given that a-$MoS_{3-x}$ is 10-fold more stable (Fig. 9), either 1) a-$MoS_{3-x}$ should present 100-fold lower intrinsic stability at equivalent HER rates to compensate for the 10-fold higher TOFs in c-$MoS_2$ or 2) c-$MoS_2$ should have a 100-fold higher density of Mo-H sites, the latter being incorrect as mentioned earlier.

Indeed, c-$MoS_2$ and $[Mo_3S_{13}]$-based catalysts presented, at low HER rates, consistently different stabilities across different HER rates to those of c-$MoS_2$, regardless of the operating conditions (Figs. S20–S22). However, the stability crossover found when comparing low and high HER rates for c-$MoS_2$ (Mo S-number ~$10^7$ vs ~$10^5$, see Figs. 5 and 8) suggests that intrinsic active site stabilities are not constant, and are highly dependent on the preferential HER and dissolution pathway. Such hypothesis is reinforced when analysing the post-mortem S-to-Mo XPS ratios after long-term testing, which indicate that near-surface sulphur species are favourably retained in c-$MoS_2$ when compared to a-$MoS_{3-x}$ (S-to-Mo ca. 1.5 vs. 0.4, see Table S4). This leads us to believe that, for c-$MoS_2$, Mo-H formation cannot solely be responsible for the HER activity. At low HER rates, given the experimental evidence of low Mo dissolution and almost neglibible S loss (Figs. 6, 8 and S17–S19), we hypothesize that unsaturated $S^{2-}$ sites are primarily involved as proton-accepting groups. At higher HER rates (Fig. 9), given the almost stoichiometric near-surface S-to-Mo ratios found in c-$MoS_2$ after long-term testing (ca. 1.5, see Table S4), we propose that $Mo^{3+}$ hydride formation (steps I' and II' in Fig. 10) would be responsible for the HER activity as well as the dissolution pathways via stoichiometric loss of Mo and S in step VI' in Fig. 10.

### Analysis of structure-activity-stability relationships

We should now evaluate the overall long-term viability of $MoS_x$ catalysts as non-noble cathodes for PEMWEs. Considering that Pt dissolution under cathodic potentials is virtually negligible[21], unless extreme voltage values are employed to induce cathodic corrosion[69,70], long-term PEMWE durability has been bound to the lifetime estimation of the state-of-the-art Ir-based nanoparticulate catalysts. Reported S-numbers for commercial $IrO_2$ (Alfa Aesar, ~$10^4$–$10^5$)[53,55], $IrO_2/TiO_2$ (Umicore, ~$10^5$)[71], and rutile $Ir_{0.2}Ru_{0.8}O_2$ catalysts (Ir: ~$10^6$, Ru: ~$10^4$)[52] in aqueous model systems (AMSs) would demand that any non-noble catalyst used to replace Pt should present comparable stabilities so as to not compromise the overall PEMWE durability (i.e. S-number $\geq 10^5$). Under such criterion, c-$MoS_2$ would seem the most suitable catalyst at low HER rates, as the S-numbers for Mo (~$10^7$) and S (~$10^6$) under HER potentials are beyond this threshold. Regardless, the high instability observed for Mo moieties upon voltage reversal/shut-down would prevent any intermittent PEMWE operation in a real device. In contrast, the stability of $[Mo_3S_{13}]$-based catalysts markedly increased up to S-numbers ~$10^5$ and ~$10^6$ under higher current densities of −10 mA $cm^{-2}$ (Figure S20-22) and −100 mA $cm^{-2}$ (Fig. 9), respectively. Hence, $MoS_x$-based PEMWEs would initially seem bound to operate under low constant current loads when using c-$MoS_2$, while high-current operation even under intermittent mode would be suited for $[Mo_3S_{13}]$-based catalysts. It is noteworthy to mention that lifetimes of $IrO_x$ and non-noble anode catalysts were dramatically extended in membrane electrode assembly (MEA) environments[72,73], which could also apply to a $[Mo_3S_{13}]$-based cathode even more than that observed after long-term H-cell measurements. For CoP, another non-noble HER catalyst, a similar trend was observed: fairly poor stability in AMSs (S-number ~$10^2$)[22] contrasted with stable long-term operation in PEMWEs (>1700 h at 1.86 A $cm^{-2}$)[6]. Indeed, minor degradation on a $MoS_x$-N-CNT cathode-based PEMWE was reported after 100 h of constant operation[7], which supports our initial assessment. Thus, a thorough stability assessment in MEA environments would be required to fully estimate the effective $MoS_x$ lifetime, beyond the scope of this report.

To the best of our knowledge, we report the first comprehensive stability assessment of $MoS_x$ catalysts by simultaneously monitoring Mo and S dissolution under relevant electrocatalytic conditions. Our work not only confirmed the key role of selective S loss in the electrochemical activation of a-$MoS_{3-x}$ and $[Mo_3S_{13}]^{2-}$ cluster-based catalysts by means of liquid (SFC-ICP-MS and H-cell) and volatile product analysis (EC-MS), besides surface species analysis (ex-situ Raman and XPS), but also the distinct stability trends across different $MoS_x$ allotropes. The lamellar-like c-$MoS_2$, structurally analogous to crystalline $MoS_2$ presents, at lower HER rates, a higher stability (10–100 fold) at the expense of lower activity versus $[Mo_3S_{13}]$-based catalysts, the latter being 10-fold more stable after long-term, high-current density operation. In addition, irrespective of the synthetic method and operating conditions, $MoS_x$ catalysts based upon the $[Mo_3S_{13}]^{2-}$ cluster backbone structure present analogous activity-stability trends. Upon mimicking intermittent operation relevant for PEMWE coupling with renewable energy inputs, we conclude that the preferential Mo loss at non-HER potentials severely compromises any $MoS_x$ cathode lifetime unless high HER rates are employed for $[Mo_3S_{13}]$-based catalysts. Based on our findings, we believe that the recently proposed $Mo^{3+}$ hydride site is likely responsible for the Mo instability observed at non-HER potentials, but cannot be unambiguously considered the universal HER active site given the stability differences observed. Although still to be fully corroborated with downstream analytics, non-noble PEMWEs based on such catalysts are recommended to be employed under constant current loads to prolong device lifetimes given the almost 5-fold higher dissolution under intermittent operation. We believe that the findings reported here will provide a better understanding of the activity-stability relationships of $MoS_2$ and their innate allotrope dependence, which have a profound impact on the design of MoS2-based catalysts and can be extended to other 2D or non-noble HER electrocatalysts.

## Methods

### Electrodeposition of cathodic $MoS_2$ and anodic $MoS_{3-x}$ thin films

**Materials.** As $MoS_x$ precursor solution, an aqueous solution of 10 mM $(NH_4)_2MoS_4$ (99.97%, Sigma-Aldrich, Germany) with 0.1 M KCl (analytical grade, Penta s.r.o., Czech Republic) was used. Before usage, the solution was filtered using a syringe filter with 0.45 μm pore size. The solution was prepared with deionized water with a resistivity of 18 MΩ cm.

**Electrodepostion.** For the electrochemical deposition of $MoS_x$ an Autolab Bipotentiostat/Galvanostat (PGSTAT302N, Metrohm, Netherlands) in three-electrode setup with a graphite counter electrode, and a Ag/AgCl 3 M KCl reference electrode was used. The deposition was performed by chronoamperometry using a glassy carbon plate (25 mm × 25 mm x 1 mm, Sigradur G, Hochtemperatur-Werkstoff GmbH, Germany) as substrate. Before the electrodeposition the glassy carbon plate was polished with $Al_2O_3$ suspension (0.05 μm) and

polishing cloth, rinsed with water, sonicated with Ethanol (96%, analytical grade, Merck KGaA, Germany) and dried with $N_2$. The potentials were $E_{substrate}$ = 0.6 V for the anodic deposition and $E_{substrate}$ = −1.3 V for the cathodic deposition. The deposition time was 10 min and the spot size was 7 mm in diameter. After the deposition, the samples were rinsed with water and dried in ambient conditions. The mass loading of the $MoS_x$ was calculated from the measured charge according to the literature reported chemical equations 1 and 2[29].

Anodic deposition: $MoS_4^{2-} \rightarrow MoS_3 + \frac{1}{8}S_8 + 2e^-$ (1)

Cathodic deposition: $MoS_4^{2-} + 2e^- + 4H^+ \rightarrow MoS_2 + 2H_2S$ (2)

For side-by-side comparison between anodic (a-$MoS_{3-x}$) and cathodic (c-$MoS_2$) electrodeposited thin films, loading was ca. 135 $\mu g_{cat}$ $cm^{-2}$. For stability benchmarking across $[Mo_3S_{13}]$-based catalysts, a-$MoS_{3-x}$ loading was set to ca. 60 $\mu g_{cat}$ $cm^{-2}$

## Physical characterization

For the physicochemical characterization of the electrodeposited $MoS_x$, scanning electron microscopy (SEM) was performed using a Tescan MIRA 3 XMU with an accelerating voltage of 20 kV. For analysis of the chemical composition, X-ray photoelectron spectroscopy (XPS) measurements with a monochromatic Al Kα X-ray source (1486.6 eV, 15 kV) were conducted using either a Kratos AXIS Supra (a-$MoS_{3-x}$ /c-$MoS_2$, power: 225 W) or a PHI Quantera II scanning X-ray microprobe ($[Mo_3S_{13}]^{2-}$ cluster catalysts, spot size: 200 μm, power: 50 W). For survey spectra acquisition, 280 eV pass energy and 1 eV step sizes were employed, whereas for high-resolution spectra these were respectively 140 eV and 0.125 eV (dwell time per step: 500 ms). All high-resolution spectra were energy-corrected to the adventitious C 1s peak set to 284.6 eV, and processed using CasaXPS (version 2.3.22PR1.0). For high-resolution spectra deconvolution, Shirley or Tougaard type backgrounds, and Functional Lorentzian (Mo 3d, LF(1,1,35,280))[74], Gaussian-Lorentzian (S 2p, GL(30)) or Lorentzian Asymmetric (LA 1, 53, 243) lineshapes were employed. Mo 3d spectra were fitted by applying a 3:2 area ratio constraint and 3.1 eV separation on the $3d_{5/2:3/2}$ spin-orbit doublets, whereas S 2p spectra were fitted by applying a 2:1 area ratio constraint and 1.18 eV separation on the $2p_{3/2:1/2}$ spin-orbit doublets. The analysis of the crystalline structure of $MoS_x$ samples was performed using an X-ray diffractometer (Rigaku 3 SmartLab, 3 kW) using Cu $K_{\alpha1}$ (λ = 0.15418 nm) radiation with Brag − Brentano geometry. The diffractogram was operated at a current of 30 mA and a voltage of 40 kV. The Raman spectroscopy measurements were performed using a confocal Raman spectroscopy system (WITEC Alpha 300 R) with a green laser (λ ≈ 532 nm).

## Ink preparation and dropcasting of $[Mo_3S_{13}]$-based electrocatalysts

The synthesis of $[Mo_3S_{13}]$-based catalysts employed in this work has been reported elsewhere[7]. The electrocatalyst inks employed in this work were prepared by dissolving 6.03 and 5.8 mg of $[Mo_3S_{13}]^{2-}$ and $Mo_3S_{13}$-N-CNT powders into an aqueous-based solution containing an 87.5/12.5 v/v ratio of ultrapure water (Merck, Milli-Q IQ 7000, 18.2 MΩ cm)/isopropanol (Merck, Emsure). A perfluorinated Nafion resin suspension (5 wt. %, Sigma Aldrich) was further added to ensure good physical binding to the working electrode, yielding a catalyst-to-ionomer weight ratio of 11/1 (final ink concentration = 5.65 g $L^{-1}$, Nafion contribution to total solid content ca. 9%). The obtained suspensions were sonicated with an ultrasonication horn (Branson, SFX 150) for 10 min at 4 s/2 s on/off pulse intervals and 40% pulse amplitude in an ice bath. Finally, multiple catalyst spots of $[Mo_3S_{13}]^{2-}$ and $Mo_3S_{13}$-NCNT were obtained by dropcasting 0.2 μL of the aforementioned inks onto a mirror-polished 5 × 5 cm glassy carbon plate (HTW, SIGRADUR), giving a loading per spot of ≈ 60−100 $\mu g_{cat}$ $cm^{-2}$. The specific diameter of the dropcasted catalyst spots (Ø ≈ 1.3 mm) was elucidated by a laser scanning microscope (Keyence, VK-X250). Before electrochemical experimentation, each catalyst spot to be tested was vertically aligned

to the SFC opening employing a computer-controlled camera and a micro-positioning stage.

## On-line inductively coupled plasma mass spectrometry

Scanning flow cell (SFC) electrochemical testing was carried out with a LabVIEW-controlled Gamry Reference 600 potentiostat (Gamry, USA), consisting of a graphitic rod counter electrode compartment (6 mm diameter, 99.995%, Sigma-Aldrich) and a double-junction Ag/AgCl reference electrode compartment (Metrohm, Switzerland; outer compartment filled with 0.1 M $HClO_4$, inner compartment with standard 3 M KCl electrolyte). Both compartments were connected to the main cell body with Tygon tubing (internal diameter: 1.02 mm). All potentials in this work are presented with respect to the reversible hydrogen electrode (RHE), after experimentally elucidating the RHE potential of the employed Ag/AgCl reference electrode using a Pt wire working electrode (0.5 mm, Premion 99.997%, Alfa Aesar) under hydrogen saturation. The V-shaped polycarbonate SFC was CNC machined in-house (CAM 4-02 Impression Gold, vhf camfacture AG, Germany), presenting an elliptical-shaped opening at the flow channels intersect, effectively providing a working electrode area of 0.033 $cm^2$.

Real time, simultaneous analysis of Mo and S dissolution from $MoS_x$-based electrocatalysts was achieved by pumping a freshly-prepared 0.1 M $HClO_4$ electrolyte (70 %, Suprapur, Merck; pH = 1) from an Ar-saturated reservoir, downstream via the V-shaped SFC channels, towards a Perkin Elmer NexION 350x inductively coupled plasma mass spectrometer (ICP-MS) connected with PTFE tubing (internal diameter: 300 μm) at a constant flow rate of ca. 195 μL·$min^{-1}$. To prevent the interference of $^{32}O_2^+$ and $^{32}NO^+$ dimers presenting high intensity at m/z = 32 of $^{32}S$, the ICP-MS instrument was operated under dynamic-reaction-cell (DRC) mode, using oxygen as the reaction gas. Employed DRC parameters were the following: cell gas A ($O_2$) = 0.5 mL $min^{-1}$, DRC Mathieu parameter a (RPa) = 0, and DRC Mathieu parameter q (RPq) = 0.25. Given the lower limit of detection (calibration curve intercept + 3σ) found for $^{50}SO^+$ dimer (ca. 36.4 ng·$L^{-1}$) compared with the $^{48}SO^+$ dimer (ca. 210 ng·$L^{-1}$), all S dissolution values are reported with respect to the $^{50}SO^+$ dimer. The ICP-MS instrument was calibrated using a five-point calibration curve obtained from standard solutions (0, 0.1, 0.5, 1 and 5 μg·$L^{-1}$) containing intentional amounts of Mo (Merck Certipur) and S (1 M $H_2SO_4$ prepared from 96% Suprapur, Merck), using 10 μg·$L^{-1}$ $^{103}Rh$ and $^{45}Sc$ as internal standards. For additional information regarding the custom setup employed, we refer to previous publications[75–77].

## Electrochemical mass spectrometry

Electrochemical mass spectrometry (EC-MS) experiments were performed on a SpectroInlets apparatus (SpectroInlets Aps, Denmark). The electrochemical setup is in a thin layer configuration. The working electrode is separated from the sampling chip by a 100 μm thick Teflon® separator. The make-up gas flown through the chip was 5.0 Helium (Messer) in all instances. All EC-MS experiments were performed in 0.1 M $HClO_4$ (Sigma-Aldrich) solution with MQ water (Merck Millipore). The cell was cleaned by thorough washing with MQ water and was stored in a covered beaker filled with distilled water to prevent contamination. Expected detection limits for $H_2$ (m/z = 2) and $H_2S$ (m/z = 34) are 10 and 175 pg $s^{-1}$ $cm^{-2}$, respectively.

First the sample working electrode disc (glassy carbon, Ø = 5 mm, a-$MoS_{3-x}$ /c-$MoS_2$ loading ca. 135 $\mu g_{cat}$ $cm^{-2}$) was installed into the electrochemical cell. The electrochemical cell was then assembled on the EC-MS setup without electrolyte. The counter electrode (Pt wire) and the reference electrode (reversible hydrogen electrode RHE, Gaskatel HydroFlex) were fitted onto the cell. The potential was set to 0 V vs. RHE and the electrolyte (0.1 M $HClO_4$) was introduced. The cell was then left at 0 V vs. RHE for 10 minutes before any electrochemical experiment was started.

## Electrochemical catalyst testing

Two main electrochemical protocols were employed to test the stability of MoS$_x$-based catalysts under HER conditions. The first one consisted of a hold at 0 V for 5 mins ([Mo$_3$S$_{13}$]-based catalysts) or 10 mins (a-MoS$_{3-x}$ and c-MoS$_2$) to resolve the dissolution signal obtained after the electrolyte contact with the WE, followed by a preconditioning cycling step, a 0 V hold to resolve the ICP-MS signal, and finally a set of symmetrical on/off HER galvanostatic holds. For the [Mo$_3$S$_{13}$]-based cluster catalysts, the preconditioning consisted of 150 cyclic voltammograms (CVs) from 0 to −0.2 V$_{RHE}$, whereas for a-MoS$_{3-x}$ /c-MoS$_2$ of 100 CVs from 0 to −0.25 V$_{RHE}$ (scan rate: 100 mV s$^{-1}$). Such preconditioning aimed to trigger the electrochemical activation of the MoS$_x$ electrocatalysts, as shown in previous reports[29,48]. The on/off HER holds were applied to evaluate catalyst activity and stability under start/stop conditions, relevant for intermittent PEMWE operation, and similar to those employed by Ledendecker et al.[21]. These consisted of six symmetrical −1 mA cm$^{-2}_{geom}$ /0 V$_{RHE}$ holds (2 mins on/off) for the [Mo$_3$S$_{13}$]-based cluster catalysts, whereas for a-MoS$_{3-x}$ /c-MoS$_2$ the duration of the 0 V$_{RHE}$ holds was extended to 5 mins to fully resolve the Mo dissolution signal in cathodic MoS$_x$.

The second protocol consisted of sequential galvanostatic holds (−1, −2, −5, −10 mA cm$^{-2}$) separated by 8 min holds at 0 V$_{RHE}$ to resolve Mo dissolution, preceded by a 10 min hold at 0 V$_{RHE}$ to resolve the dissolution signal obtained after the electrolyte contact with the WE. All error bars presented in SFC-ICP-MS experiments stem from the standard deviation of three independent measurements.

## H-cell stability measurements

Electrochemical long-term stability tests were conducted in commercial H-cells (Pine Research). Each compartment was filled with 30 mL of 0.1 M HClO$_4$, prepared from 70% HClO$_4$ (Suprapur, Merck). Prior to measurement, the electrolyte was saturated with argon for 30 minutes and maintained above the solution throughout the experiment. The setup included a working electrode of electrodeposited MoS$_x$ on glassy carbon (SIGRADUR G, HTW) and an Ag/AgCl reference electrode (Metrohm, Germany) in one compartment. A glassy carbon counter electrode (SIGRADUR G, HTW) in the opposite compartment minimized Mo ion redeposition. A magnetic stirrer at 800 RPM ensured consistent convection, preventing local concentration gradients. Control of potential and current was achieved using a VSP-300 potentiostat (Biologic). The study employed two distinct protocols: the first involved a consistent −100 mA cm$^{-2}$ galvanostatic hold for 5 h to evaluate stability. The second protocol consisted of alternating −100 mA cm$^{-2}$ pulses with 0 V$_{RHE}$, each for 2 min, continuing for a total duration of 5 h. In addition, a 30 min 0 V$_{RHE}$ potentiostatic hold was applied before and after each protocol. Ohmic resistance was determined by impedance spectroscopy, specifically at the high-frequency interception of the Re(Z) axis, to fully compensate for iR drop during the protocols. Sample aliquots (V = 500 μL) were taken at predetermined intervals, and analyzed offline using ICP-MS. Consistency in the total volume of electrolyte, maintained at 30 mL in both compartments, was ensured throughout the experiment. All error bars presented stem from the standard deviation of two independent measurements.

## Data availability

The authors declare that the data supporting the findings of this study are available within the paper and its Supplementary Information files. The processed electrochemical, ICP-MS, EC-MS and XPS datasets presented in the main manuscript are available at Jülich DATA database (https://doi.org/10.26165/JUELICH-DATA/NXROSN). Additional data can be provided from authors upon request.

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

## Acknowledgements

D.E.-L., M.Z., and S.C. gratefully acknowledge the DFG for financial support within the grant CH 1763/3-1 as a part of the Priority Program SPP 2080 "Catalysts and reactors under dynamic conditions for energy storage and conversion". D.E.-L. would also like to thank Dr. Mario Löffler and Dr. Florian Speck for XPS acquisition. M.P. acknowledges the financial support of the Grant Agency of the Czech Republic (EXPRO: 19-26896X). C.I., M.P. and K.N. acknowledges the financial support by the European Union's Horizon 2020 Research and Innovation Program under the Marie Skłodowska-Curie grant agreement no. 888797 and Czech-NanoLab project LM2023051 funded by MEYS CR for the financial support of the measurements and sample characterisation at CEITEC Nano Research Infrastructure. N.M., P.J. and N.H. would like to thank the financial support from the Slovenian Research Agency through the research programs/projects I0-0003, P2-0393, P2-0421, N2-0248, N2-0257 and N2-0155. In addition, they would like to acknowledge NATO Science for Peace and Security Program under Grant G5729.

## Author contributions

D.E.-L. and S.C. conceived the project and design of experiments. D.E.-L. performed sample dropcasting, online ICP-MS measurements, physical characterization (XPS), data analysis and was responsible for manuscript write-up. M.Z. performed online ICP-MS and long-term H-cell measurements, including data analysis. C.I., K.N. and C.V.P. conducted the sample synthesis and physical characterization (SEM, XPS). N.M. and P.J. performed the EC-MS measurements. S.C., S.T., N.H. and M.P. were responsible for project supervision, funding acquisition, manuscript reviewing and editing. All authors discussed the results, revised the manuscript and gave their approval prior to submission.

## Funding

## Competing interests

The authors declare no competing interests
