## [Peer Review File · Nature Communications]

Allotrope-dependent activity-stability relationships of molybdenum sulfide hydrogen evolution electrocatalystsREVIEWER COMMENTS

Reviewer #1 (Remarks to the Author):

Lopez et al. have robustly assessed the stability of the model MoS₂ catalysts with a scanning flow cell coupled with downstream analytics (ICP-MS). Herein, two types of MoS₂ are studied, lamellar-like MoS₂ and cluster-like amorphous MoS_{3-x}. They have identified that lamellar-like MoS₂ is highly unstable under open circuit conditions whereas cluster-like amorphous MoS_{3-x} instability is induced by S loss and the generation of undercoordinated Mo sites. Though the methods and major goals of this work are novel and interesting, the quality of this work might not be suitable for publishing at Nature Communications.

1. The reviewer does have some concerns about why the authors chose the two types of MoS₂, rather than the crystalline phase transitional 2H-MoS₂ or 1T-MoS₂, which has a well-known structure. One question with the two types of MoS₂ is do they really have the structure as predicted in Scheme 1. It is well known that amorphous phase compounds do not have a repeat pattern of crystal structure. Thus, it is hard to use a structure to predict its property.

2. Further, the tracked structure changes by flow cell coupled with downstream analytics should also couple with in-situ electrochemical performance and correlated structural analysis. Without a well-defined structural analysis, it is hard to agree with the conclusion that has been drawn in the paper. However, amorphous phases do have a limitation for defined structure analysis.

3. Lastly, ICP-MS is a well-known technique for metal analysis. How it can accurately measure the S amount?

Reviewer #2 (Remarks to the Author):

The paper entitled « Phase-dependent activity-stability relationships of molybdenum sulfide hydrogen evolution electrocatalysts» submitted to Nature Communications focuses on the stability of MoS₂-based catalysts for the hydrogen evolution reaction in acidic conditions. Despite its low turnover frequency, it is believed that this class of non-noble can replace platinum at the cathode of water electrolyzers. However, as stated by the authors, their durability is often overlooked at the expense of initial activity considerations. Online ICP-MS measurements are conducted to monitor Mo and S dissolution. This allows to get mechanistic insights and unique results on the stability of MoS₂ catalysts. This work is well conducted and of high interest for the electrocatalysis community. I recommend publication after addressing the following comments in a revised version of the manuscript. By doing so, the conclusions will certainly be strengthened.

1-The title of the present paper is “Phase-dependent activity-stability relationships” but the different phases of MoS₂ (2H or 1T phases) are not discussed. In my view, this study compares the stability trends

of crystalline and amorphous phases. Since the works of Jaramillo's group beginning in the 2000s, it is well known that amorphous MoS₂ are much active but less stable for HER. The authors should clarify this point on the definition of "phase" and discuss the existing literature on the topic.

2- Comparing stability without knowledge of the density of active sites is complicated. Even if the loading of MoS₂ on the electrode is similar, the density of active sites can differ and this will influence the conclusions, in particular the amount of Mo and S dissolved (Figure 3 and 5). Can the authors comment on that?

3-The HER activities of the a and c MoS₂ are very low in regards of literature: e.g. the catalysts used in the present paper do not achieve - 10 mA cm⁻²geom (Figure 1), which is the figure of merit for HER catalysts. With such low HER activity, drawing general conclusions for this class of catalysts is highly speculative.... In the same vein, how can the authors draw general conclusions from such a mild durability test? In my view, six consecutive start stop cycles from -1 mA cm⁻²geom to 0 VRHE are not representative of PEMWE application. How generalizable are the results obtained in this study?

4- It would have been interesting to quantify the amount of S and Mo dissolved with respect to their initial loading on the electrode (as it has been done in the recent paper of Zavala and al. ACS Catal. 2023, 13, 1221–1229).

Reviewer #3 (Remarks to the Author):

Since MoS₂ is regarded as a competitive to replace platinum in PEMWEs, the authors investigated the stability of MoS₂ cathodes with different crystal structures (a-MoS₂ and c-MoS₂) under acidic conditions. This is a pretty new perspective in researching the over-investigated electrocatalysts and they found that a-MoS₂ possessed a high activity while c-MoS₂ had better stability at the HER operating potential. Furthermore, the cathodic dissolution pathways and HER mechanism were detailed discussed. They concluded that c-MoS₂ presented the best trade-off between activity and stability if PEMWEs were to operate under constant load. However, the most doubtable points are the not excellent activity of c-MoS₂ according to the LSV data and not realistic working condition (up to 10 mA/cm²). To the best of our knowledge, many catalysts with high activity and high stability under acidic conditions have been designed [Sci. Adv. 9, eadh2285 (2023)], [Nat. Commun. 13, 7784 (2022)], which are even much better than a-MoS₂. Thus, I would like to recommend a major revision and improvement before further consideration of this manuscript. Here are the detailed comments:

1. This work only studied the ion dissolution of the catalyst under the small current density (up to 10 mA cm⁻²). While a high current density, at least larger than 100 mA cm⁻², is required to meet the practical operating condition. Under such a high current density, the failure mechanism of the catalyst may change. Therefore, I think the stability at high current densities is also noteworthy to be investigated.
2. The long-term stability of these two electrodes needs to be supplemented to quantify the effect of sulfur dissolution on the electrode stability.
3. More characterizations, such as XRD, are required to confirm that electrodes with two different crystal

structures have been successfully prepared.

4. In addition to EC-MS and XPS, SEM and other characterizations needs to be conducted after the stability test to rule out other factors besides dissolution that cause electrode failure.

5. Sulfur dissolution of c-MoS₂ under “start-up/shutdown” HER holds (Figure 4) demonstrates that a large amount of sulfur dissolution occurs in the first “start-up/shutdown” cycle (around 1600 s) while remaining at a low value in subsequent cycles. Please explain this phenomenon.

6. Some formatting mistakes should be corrected, such as “a-MoS_{3-x} and c-MoS₂” in line 82. Please double-check the manuscript.

Response to reviewers

Reviewer #1 (Remarks to the Author):

Lopez et al. have robustly assessed the stability of the model MoS₂ catalysts with a scanning flow cell coupled with downstream analytics (ICP-MS). Herein, two types of MoS₂ are studied, lamellar-like MoS₂ and cluster-like amorphous MoS_{3-x}. They have identified that lamellar-like MoS₂ is highly unstable under open circuit conditions whereas cluster-like amorphous MoS_{3-x} instability is induced by S loss and the generation of undercoordinated Mo sites. Though the methods and major goals of this work are novel and interesting, the quality of this work might not be suitable for publishing at Nature Communications.

We would like to thank the reviewer for acknowledging the novelty and interest of our manuscript, and we hope that our detailed response and additional measurements aimed at resolving the concerns raised by the reviewers can be convincing to find our work suitable for being published in Nature Communications.

1. The reviewer does have some concerns about why the authors chose the two types of MoS₂, rather than the crystalline phase transitional 2H-MoS₂ or 1T-MoS₂, which has a well-known structure. One question with the two types of MoS₂ is do they really have the structure as predicted in Scheme 1. It is well known that amorphous phase compounds do not have a repeat pattern of crystal structure. Thus, it is hard to use a structure to predict its property.

We understand the reviewer's views regarding the choice 2H and 1T phases having a well-known crystalline structure, and we agree that such systems would provide very interesting insights. However, we must remark that all recent efforts aimed towards MoS₂-based catalyst implementation in PEMWE devices have almost exclusively devoted to the use of the less crystalline yet more active MoS_x allotropes, namely the cluster-based [Mo₃S₁₃]²⁻ and electrodeposited MoS_x as reported in recent publications (see Catalysts **2018**, *8*, 657; ACS Sustainable Chem. Eng. 2023, *11*, 20, 7641–7654). One of the main advantages of using such catalyst types is that the synthetic routes generally employed to prepare them allow a very straightforward implementation onto gas diffusion electrodes/porous transport layers suitable for PEMWEs. For electrodeposited MoS_x, GDEs/PTLs can be directly employed as working electrodes, allowing a scalable cathode decoration (see Ng et al., ChemSusChem 2015, *8*, 3512). Indeed, one of the core points addressed in our manuscript is the assessment of the activity-stability relationships of such materials under operating conditions relevant for PEMWE applications by carefully understanding how dissolution processes are triggered in such materials. This, we believe, is of high importance to fully implement non-noble cathode catalysts in commercial PEMWE devices. Although insightful, the study of crystalline 2H and 1T allotropes of MoS₂ would require a dedicated, self-contained study beyond the scope of this work.

With regards to the reviewer's concerns correlating amorphous MoS_x catalysts structure and properties (i.e. activity and stability), we acknowledge that such materials do not possess long-range periodical order as crystalline materials. This is showcased by the broad XRD patterns

obtained for a-MoS_{3-x} and c-MoS₂ dominated by the glassy carbon backing electrode contribution, shown in Figure S25, S26 and S28 in the revised Supporting Information. To exclude the glassy carbon backing electrode influence in the XRD acquisition, pristine a-MoS_{3-x} and c-MoS₂ films were prepared on a Si substrate to obtain additional diffractograms (Figure R1). When compared with the ICDD database card of crystalline MoS₂, it is clear indeed that both thin films provide a very broad diffractogram dominated by the underlying Si substrate.

Figure R1. Left) XRD diffractograms of cathodically (blue) and anodically (green) electrodeposited MoS_x on Si substrate. Right) Reference tabulated XRD data for crystalline MoS₂ (ICDD card no. 37-1692).

However, we must point out that a-MoS_{3-x} and c-MoS₂ have been thoroughly studied during the last decade with a plethora of characterization techniques (for representative studies, we refer the reviewer to references 29-34 in the revised manuscript, as well as *ChemSusChem* 2019, 12, 483-4389 and *ACS Catal.* 2019, 9, 3, 2568–2579, among other works). Indeed, recent HAADF-STEM imaging could resolve the coordination polymer structure present in a-MoS_{3-x} (see Figure 3 in Tran et al., *Nat Mater* 2016, **15**, 640-646). Therefore, we believe that there is enough bibliographic evidence which supports that, while not presenting long-range crystalline order, amorphous MoS_x materials can be consistently ascribed to specific structures and therefore correlate these to specific properties, in this case their electrochemical activity and stability toward the hydrogen evolution reaction. To further reinforce this statement, our work remarkably showcases that despite of using different synthetic routes, all materials studied containing the [Mo₃S₁₃]²⁻ motif present virtually identical activity-stability relationships when

directly assessing these with the S-number metric. Thus, our work would indeed point towards the fact that the intrinsic structure motifs yield analogous activity-stability relationships.

2. Further, the tracked structure changes by flow cell coupled with downstream analytics should also couple with in-situ electrochemical performance and correlated structural analysis. Without a well-defined structural analysis, it is hard to agree with the conclusion that has been drawn in the paper. However, amorphous phases do have a limitation for defined structure analysis.

We thank the reviewer for his comment. As well pointed out by the reviewer, the lack of long range periodic order in a-MoS_{3-x} and c-MoS₂ precludes most efforts when performing correlated structural analysis. From the reviewer's suggestion, however, we gather that he/she would be interested in having additional insights from techniques such as identical location transmission electron microscopy (IL-TEM) or liquid cell electrochemical transmission electron microscopy (EC-TEM). Although such techniques are extremely powerful and provide a high degree of structural information, they are normally not available in most international laboratories and even if employed would have the extreme challenge to resolve the small motifs present in, for example, the a-MoS_{3-x} coordination polymer structure which are in the nanometer-to-angstrom scale (for representative HAADF-STEM of the cluster structure, see reference 32). Therefore, these would require a dedicated, self-contained study using either IL-TEM or EC-TEM, beyond the scope of this work.

During the reviewing timespan of our manuscript we identified a publication which partially targets the question posed by the reviewer devoted to Re-doped MoS₂ catalysts (*ACS Appl. Mater. Interfaces* 2023, 15, 40, 46895–46901, included in the revised manuscript as reference 20) as well as a pre-print dated in mid-September 2023 (<https://physics.paperswithcode.com/paper/operando-insights-on-the-degradation>) from Christina Scheu's group. Both studies provide insights on the Re-doped MoS₂ lamellar morphology but they do not tackle two important aspects investigated at length in our study: 1) the role of S loss in activity-stability relationships (as we do here by tracking online both liquid and volatile sulphur products under hydrogen evolution potentials) and 2) its implications in the hydrogen evolution mechanism of amorphous MoS_x electrocatalysts. We believe that the results compiled in our manuscript provide enough evidence for the aforementioned, which we consider will provide very valuable information to the non-noble PEMWE scientific community.

To address the reviewer's concerns regarding structural analysis, we have performed additional ex-situ SEM, XRD, XPS and Raman measurements before and after long-term stability testing (5 hours at constant current densities of -100 mA cm⁻² or start/up-shut-down between -100 mA cm⁻² and 0V vs. RHE) to monitor the morphological and surface composition change. For ease of convenience, the resulting figures (see Figures S25-S29 in Supporting Information) are included right after the detailed response, and the information contained here is paraphrased from what is present in the revised manuscript (see responses to question 3 from Reviewer#2 and question 1/2 from Reviewer #3).

Both SEM micrographs (A-C) and XRD diffractograms (D-F) present no major modification before and after long-term HER testing, indicating that the stability trends are not hampered by the mechanical stability or change in crystallinity of both material sets. (see Figures S26 and S28) The Raman spectra present characteristic traits of each allotrope studied. For *c*-MoS₂ (F, Figure S25), which presents a lamellar-like structure similar to that of crystalline MoS₂, two Raman modes are present: namely E¹_{2g} and A_{1g} at 380 and 405 cm⁻¹, respectively. These modes are in excellent agreement with those found for crystalline MoS₂ (see *ACS Nano* 2016, 10, 2, 1948–1953). In the case of *a*-MoS_{3-x} (E, Figure S25), the pristine sample presenting characteristic Raman modes at 450, 520 and 550 cm⁻¹ arising from apical S²⁻, bridging and terminal S²⁻ from the cluster-based [Mo₃S₁₃]²⁻ structure, among Raman modes arising from weaker Mo-S bond in the 200-400 cm⁻¹ region (see *Nat Mater* 2016, **15**, 640-646). After long-term testing, Raman measurements showcased a clear loss of the distinct E¹_{2g} and A_{1g} Raman modes in *c*-MoS₂ (see G-I Figure S26). In addition, long-term tested *a*-MoS_{3-x} presents the loss of S²⁻_{bridg}/S²⁻_{term}/S²⁻_{ap} Raman modes (ca. 550, 520 and 450 cm⁻¹) and the well-reported appearance of both a band at 430 cm⁻¹ (resembling A_{1g}) and MoO_x-related Raman features (broad band at 800-1000 cm⁻¹), corroborate the structural transformation to the MoS_{2-x} structure via S loss and undercoordinated Mo site generation (see G-I Figure S28). In addition, it is well reported in the literature that the S 2p XPS region is a footprint for both anodically and cathodically electrodeposited MoS_x materials (see *ACS Catal.* 2013, 3, 2002-2011), arising from differences in the relative abundance of S²⁻ and S²⁻ moieties in their structure, clearly shown in our measurements. XPS measurements once more showcase the modification in the relative S²⁻_{term}/S²⁻_{unsat} ratio towards a higher presence of S²⁻_{term}/S²⁻_{unsat} ligands for *a*-MoS_{3-x} characteristic of the phase transformation to MoS_{2-x} under cathodic potential (Figure S29), in line with previous works (see *ACS Catal.* 2019, 9, 2368–2380, *ACS Catal.* 2016, 6, 861–867 and *ACS Catal.* 2013, 3, 2002–2011) also showcased by the lower S-to-Mo near-surface ratios (from ca. 4.3 in the pristine material to 0.3 after long-term HER testing, see Table S4 in the revised manuscript). For *c*-MoS₂, the 10-fold lower stability observed is directly linked to the increased presence of oxidized Mo⁵⁺O_xS_y:Mo⁶⁺ species at the expense of Mo⁴⁺ from MoS₂ (ca. 9:42 at. % after intermittent operation, see Figure S27).

These are results gathered post-mortem and without spatial correlation, and we acknowledge that these do not provide the correlative structural information wished by the reviewer but do provide an insightful view of the structural changes within a fairly small sample domain (catalyst spot diameter = 6 mm) at very high hydrogen evolution rates (loading-normalized currents of -741 mA mg_{cat}⁻¹), which valuable and corroborate the stability trends observed in this work.

Figure S25: Physicochemical characterization of the pristine anodically (A, C, E, G) and cathodically (B, D, F, H) electrodeposited MoS_x thin films employed for long-term stability testing. A, B) Show the scanning electron micrographs, C, D) the XRD diffractograms including the glassy carbon substrate (GC) as reference for the backing electrode contribution, E, F) the Raman spectra, and G, H) the XPS survey spectra.

Figure S26: Physicochemical characterization of cathodically electrodeposited MoS_x thin films employed for long-term stability testing: pristine (A, D, G, J), after 5 hours constant galvanostatic hold at -100 mA cm^{-2} (B, E, H, K) and after 5 hours undergoing start-up/shutdown HER holds alternating from -100 mA cm^{-2} to 0 V vs. RHE (C, F, I, L). A-C) Show the scanning electron micrographs, D-F) the XRD diffractograms including the glassy carbon substrate (GC)

as reference for the backing electrode contribution, G-I) the Raman spectra, and J-L) the XPS survey spectra.

Figure S27. High-resolution Mo 3d (top) and S 2p (bottom) XPS spectra of pristine c-MoS₂ thin films before electrochemistry (I, left), after 5 hours constant galvanostatic hold at -100 mA cm⁻² (II, center), and after 5 hours undergoing start-up/shutdown HER holds alternating from -100 mA cm⁻² to 0 V vs. RHE (III, right). Labels: cumulative peak fit (black), Mo⁴⁺ 3d_{5/2:3/2} (light blue), Mo^aO_bS_c 3d_{5/2:3/2} (blue), Mo⁶⁺ 3d_{5/2:3/2} (dark blue), S 2p_{3/2:1/2} (S²⁻, yellow), S 2p_{3/2:1/2} (S₂²⁻, orange) and S 2p_{3/2:1/2} (SO_x^{y-}, red).

Figure S28: Physicochemical characterization of anodically electrodeposited MoS_x thin films employed for long-term stability testing: pristine (A, D, G, J), after 5 hours constant galvanostatic hold at -100 mA cm^{-2} (B, E, H, K) and after 5 hours undergoing start-up/shutdown HER holds alternating from -100 mA cm^{-2} to 0 V vs. RHE (C, F, I, L). A-C) Show the scanning electron micrographs, D-F) the XRD diffractograms including the glassy carbon substrate (GC) as reference for the backing electrode contribution, G-I) the Raman spectra, and J-L) the XPS survey spectra.

Figure S29. High-resolution Mo 3d (top) and S 2p (bottom) XPS spectra of pristine a-MoS_{3-x} thin films before electrochemistry (I, left), after 5 hours constant galvanostatic hold at -100 mA cm⁻² (II, center), and after 5 hours undergoing start-up/shutdown HER holds alternating from -100 mA cm⁻² to 0 V vs. RHE (III, right). Labels: cumulative peak fit (black), Mo⁴⁺ 3d_{5/2:3/2} (light blue), Mo^aO_bS_c 3d_{5/2:3/2} (blue), Mo⁶⁺ 3d_{5/2:3/2} (dark blue), S 2p_{3/2:1/2} (S²⁻, yellow), S 2p_{3/2:1/2} (S₂²⁻, orange) and S 2p_{3/2:1/2} (SO_x^{y-}, red).

3. Lastly, ICP-MS is a well-known technique for metal analysis. How it can accurately measure the S amount?

We thank the reviewer for the query on sulphur quantification using ICP-MS. While ICP-MS is classically employed for quantification of high atomic number elements with extremely high sensitivities (sub-ppb in some instances), the technique can also be employed to quantify lower atomic number elements under specific operation parameters. In the case of sulphur, the most abundant isotope ³²S presents multiple m/z interferences with oxygen (¹⁶O₂⁺) and nitrogen dimers (primarily ¹⁴N¹⁸O⁺ and ¹⁵N¹⁶O¹H⁺), which prevent the direct quantification of m/z = 32 in the standard ICP-MS operation mode. However, sulphur can be quantified when reacting with oxygen at the so-called dynamic reaction cell (DRC) to form the dimers ⁵⁰SO⁺ and ⁴⁸SO⁺ prior to entrance to the mass quadrupole, as reported by Bandura et al. in a dedicated study (*Anal. Chem.* 2002, 74, 1497-1502). Quantification of the higher mass-to-charge dimers at m/z = 50 and 48 therefore avoids the aforementioned sulphur interferences with oxygen and nitrogen. Our online ICP-MS study implemented such approach by optimizing the oxygen reaction gas flow to maximize ⁵⁰SO⁺ and ⁴⁸SO⁺ ion signal versus the blank electrolyte. The optimized ICP-MS parameters in DRC mode were already reported in the manuscript, but for ease of convenience we have plotted the resulting calibration curves from standard solutions containing known amounts of sulphur (calibration range: 0.1 to 5 ug L⁻¹) for the dimers ⁵⁰SO⁺ and ⁴⁸SO⁺, see Figure R2.

Figure R2. Representative ICP-MS calibration curves obtained for $m/z = 48$ ($^{48}\text{SO}^+$, left) and $m/z = 50$ ($^{50}\text{SO}^+$, right).

Reviewer #2 (Remarks to the Author):

The paper entitled « Phase-dependent activity-stability relationships of molybdenum sulfide hydrogen evolution electrocatalysts» submitted to Nature Communications focuses on the stability of MoS₂-based catalysts for the hydrogen evolution reaction in acidic conditions. Despite its low turnover frequency, it is believed that this class of non-noble can replace platinum at the cathode of water electrolyzers. However, as stated by the authors, their durability is often underlooked at the expense of initial activity considerations. Online ICP-MS measurements are conducted to monitor Mo and S dissolution. This allows to get mechanistic insights and unique results on the stability of MoS₂ catalysts. This work is well conducted and of high interest for the electrocatalysis community. I recommend publication after addressing the following comments in a revised version of the manuscript. By doing so, the conclusions will certainly be strengthened.

We would like to thank the reviewer for the very positive feedback provided and for the insightful comments. We will address the reviewer's comments point by point below.

1-The title of the present paper is "Phase-dependent activity-stability relationships" but the different phases of MoS₂ (2H or 1T phases) are not discussed. In my view, this study compares the stability trends of crystalline and amorphous phases. Since the works of Jaramillo's group beginning in the 2000s, it is well known that amorphous MoS₂ are much active but less stable for HER. The authors should clarify this point on the definition of "phase" and discuss the existing literature on the topic.

We thank the reviewer for pointing out the potential discrepancy between the title and content of the manuscript. MoS₂ materials present multiple phases which yield different crystalline structures and coordination chemistry, such as the already named 2H and 1T phases. These namely differ by their structure (trigonal prismatic for the 2H phase and octahedral for the 1T phase) as well as the stacking sequence: the number prefix corresponds to the number of S-Mo-S layers involved in the periodical stacking sequence. However, perhaps the most accurate and inclusive terminology for what we aimed to convey in our work would be 'allotrope': under

such umbrella structurally-distinct MoS₂ materials with different stoichiometry or crystallinity can be included, as factually speaking neither a-MoS_{3-x} nor c-MoS₂ can be classified as crystalline materials. Indeed, while cathodically electrodeposited MoS₂ resembles the layered structure of 2H-MoS₂, it is not a purely crystalline material and therefore would not be suited to be described as one. Given the concerns raised by the reviewer concerning the concept of ‘phase’ we modified the title of the manuscript to ‘*Allotrope-dependent activity-stability relationships of molybdenum sulfide hydrogen evolution electrocatalysts*’ in the revised manuscript, as well as employing the term ‘phase’ in the manuscript only when it refers to crystalline MoS₂ allotropes. Following up the reviewer’s suggestion, we have included the definition of phase and allotrope in the Introduction of the revised manuscript, kept brief due to the word limit constraints imposed by the journal format, which prevents us from extending the MoS₂ discussion in exchange for a deeper, more insightful analysis of the results presented.

‘While a plethora of works reported detailed insights on the influence of crystalline phase (i.e. 1T, 2H and 3R, presenting different Mo-S atomic arrangements and stacking across a long-range periodical structure)^{10, 11, 12}, degree of crystallinity/disorder^{13, 14, 15} and edge-to-basal plane ratio^{16, 17, 18, 19} in the HER electrocatalytic activity, the intrinsic stability of MoS₂ under HER operating conditions is rarely (if ever) assessed besides indirect electrochemical metrics. First evaluated on crystalline MoS₂²⁰ and more recently on [Mo₃S₁₃]²⁻ cluster-based amorphous molybdenum sulfide MoS_{3-x}⁷ allotrope (presenting distinct Mo-S atomic arrangement but no long-range periodical order found in crystalline structures), a stark stability difference was observed between hydrogen-evolving and non-HER conditions’

2- Comparing stability without knowledge of the density of active sites is complicated. Even if the loading of MoS₂ on the electrode is similar, the density of active sites can differ and this will influence the conclusions, in particular the amount of Mo and S dissolved (Figure 3 and 5). Can the authors comment on that?

We thank the reviewer for his interest in the assessment of the active site density. It is well-known in the literature that, unlike platinum-group metals where multiple electrochemical methods can be employed for active site estimation (hydrogen and metal underpotential deposition, to name a few), MoS₂ does not present any specific adsorption (neither hydrogen nor metal) which can be directly related to the electrochemically active surface area. In addition, the nature of the active sites is a long-standing matter of debate in the MoS₂ community. This was already mentioned in the previous manuscript version when discussing the HER mechanism (see text below).

‘With regards to the HER active sites, recent experimental evidence from Bau et al. has directly confirmed the formation of a Mo³⁺ hydride under HER potentials in a-MoS_{3-x} using electron paramagnetic resonance in organic media, and indirectly correlated its abundance across different MoS₂ materials with the intensity of the reductive pre-peak responsible for S loss³⁸. Such Mo-H state, previously proposed by theoretical³⁶ and experimental works^{31, 37} as the HER

active site in MoS₂ materials, were recently suggested to also be universally responsible for all Mo-based HER electrocatalysis³⁹. This evidence would be in direct opposition with substantial reports which correlate HER activities with Mo-edge site length¹⁶, higher presence of proton-accepting S moieties⁶⁷ or improved phase-dependent hydrogen binding energies²⁵.

The best way to estimate the active site density would be to probe a well-defined, defect-free MoS₂ surface with the aimed crystalline structure (similarly to what was reported by Li et al. *ACS Catal.* 2015, 5, 448–455) and then extend the results to the tested materials. However, the MoS_x materials tested here present no long-range periodical order and once crystallized present modified S-to-Mo stoichiometries (as is the case of a-MoS_{3-x}, see *J. Phys. Chem. C* 2016, 120, 50, 28789–28794) and therefore cannot be assessed with this approach. Indeed, the presence of different hydrogen accepting sites after electrochemical conditioning for a-MoS_{3-x} (see *Nat Mater* 2016, 15, 640-646, *ACS Catalysis* 2016, 6, 861-867 and *Nature Catalysis* 2022, 5, 397-404) and c-MoS₂ (*Nat Commun* 2017, 8, 15113) makes the task of quantifying the density of active sites extremely challenging. Therefore, given that we cannot control the density of active sites as these virtually change under HER operation, we employed identical catalyst loadings and hydrogen evolution rates by using galvanostatic holds of identical magnitude for all catalyst studies, much more reliable when assessing activity-stability relationships. Proof of the success of this approach is that all cluster-based [Mo₃S₁₃]²⁻ catalysts studies here present analogous S-numbers regardless of loading-normalized current densities as the building motif for all materials is the [Mo₃S₁₃]²⁻ cluster.

Knowing the intrinsic limitations of the MoS₂ materials and the S-number metric, which assumes 100% Faradaic efficiencies for the gas-evolving reaction of interest (in this case the HER), we proceeded to discuss the stability results in terms of active site density in the framework of the Mo³⁺ hydride involvement in the HER mechanism in the revised manuscript as follows.

'Based on the Mo-H activity universality hypothesis, HER activity in Mo-based electrocatalysts is dictated by the amount of Mo hydride sites, if these have the same chemical environment. The Mo⁴⁺/Mo³⁺ reduction onset potential was indeed attributed to being the universal HER activity descriptor for Mo-based catalysts,⁴⁰ which is in line with the almost identical cathodic Mo dissolution onset potentials observed for all MoS_x catalysts (Figures S2, S4). This would initially indicate that the active sites have the same chemical environment. At equivalent chemical environments, the stability of all MoS_x catalysts should be comparable under identical HER rates (imposed in our start-up/shut-down stress tests), as the intrinsic stability of the Mo-H species would dictate the Mo stability trends. In other words, all Mo-based catalysts should have the same Mo S-numbers under identical mass-normalized HER current densities regardless of the amount of active sites if Mo/S dissolution is solely caused by HER. However, the different Mo S-numbers obtained for c-MoS₂ compared to a-MoS_{3-x} at low and high HER rates (100-fold higher and 10-fold lower, see Figures 5 and 8) would contradict this hypothesis:

either Mo^{3+} hydride sites would be (de)stabilized (i.e. different intrinsic activity) or an alternative HER pathway would be involved.

The intrinsic MoS_x active site activity should be therefore discussed. In order to have the same amount of Mo-H as *a*- MoS_{3-x} , *c*- MoS_2 should dissolve the same amount of S at equivalent HER rates. This is not the case, given the 100-fold lower S dissolution in *c*- MoS_2 under HER potentials at low HER rates (Figure 5). At equivalent HER rates, higher turnover frequencies per available Mo-H site would then be required. Indeed, a previous study on temperature-dependent MoS_x activity reported 10-fold higher turnover frequencies (TOF) for polycrystalline MoS_2 than amorphous MoS_{3-x} despite of higher Tafel slopes¹⁴. While having a higher number of active sites, the poorly-conducting yet more active *a*- MoS_{3-x} presents lower TOFs. In this work we observed that Mo centers in *a*- MoS_{3-x} are 100-fold more unstable compared with *c*- MoS_2 at low HER rates (Figure 5), while retaining a higher HER activity (Figure S8 ESI). This would require, at least, a 1000-fold higher density of Mo-H active sites in *a*- MoS_{3-x} to compensate for the performance losses (i.e. Mo loss) if Mo sites were only responsible for the HER activity. At high HER rates, given that *a*- MoS_{3-x} is 10-fold more stable (Figure 8), either 1) *a*- MoS_{3-x} should present 100-fold lower intrinsic stability at equivalent HER rates to compensate for the 10-fold higher TOFs in *c*- MoS_2 or 2) *c*- MoS_2 should have a 100-fold higher density of Mo-H sites, the latter being incorrect as mentioned earlier.

Indeed, *c*- MoS_2 and $[\text{Mo}_3\text{S}_{13}]$ -based catalysts presented, at low HER rates, consistently different stabilities across different HER rates to those of *c*- MoS_2 , regardless of the operating conditions (Figures S20-S22). However, the stability crossover found when comparing low and high HER rates for *c*- MoS_2 (Mo S-number $\sim 10^7$ vs $\sim 10^5$, see Figure 5 and 8) suggests that intrinsic active site stabilities are not constant, and are highly dependent on the preferential HER and dissolution pathway. Such hypothesis is reinforced when analysing the post-mortem S-to-Mo XPS ratios after long-term testing, which indicate that near-surface sulphur species are favourably retained in *c*- MoS_2 when compared to *a*- MoS_{3-x} (S-to-Mo ca. 1.5 vs. 0.4, see Table S4). This leads us to believe that, for *c*- MoS_2 , Mo-H formation cannot solely be responsible for the HER activity. At low HER rates, given the experimental evidence of low Mo dissolution and almost negligible S loss (Figures 5, 7 and S17-S19), we hypothesize that unsaturated S^{2-} sites are primarily involved as proton-accepting groups. At higher HER rates (Figure 8), given the almost stoichiometric near-surface S-to-Mo ratios found in *c*- MoS_2 after long-term testing (ca. 1.5, see Table S4), we propose that Mo^{3+} hydride formation (steps I' and II' in Scheme 2) would be responsible for the HER activity as well as the dissolution pathways via stoichiometric loss of Mo and S in step VI' in Scheme 2.'

3-The HER activities of the a and c MoS₂ are very low in regards of literature: e.g. the catalysts used in the present paper do not achieve -10 mA cm⁻²geom (Figure 1), which is the figure of merit for HER catalysts. With such low HER activity, drawing general conclusions for this class of catalysts is highly speculative.... In the same vein, how can the authors draw general conclusions from such a mild durability test? In my view, six consecutive start stop cycles from -1 mA cm⁻²geom to 0 VRHE are not representative of PEMWE application. How generalizable are the results obtained in this study?

We thank the reviewer for this concerns regarding the validity and transferability of our results to PEMWE-relevant applications. Although the geometric current densities employed in our online ICP-MS studies are small, we should point out that the loadings in the catalyst spots tested are very small (ca. 60-100 $\mu\text{g}_{\text{cat}} \text{cm}^{-2}$) and therefore the mass-normalized current densities are notable: for -1 mA cm^{-2} this corresponds to ca. 10-17 mA $\text{mg}_{\text{cat}}^{-1}$, whereas for -5 mA cm^{-2} this corresponds to ca. 50-83 mA $\text{mg}_{\text{cat}}^{-1}$. However, these are small compared to the mass-normalized currents which non-noble MoS_x-based PEMWE cathodes will undergo at 1 A cm^{-2} , which range between 333 and 1000 mA $\text{mg}_{\text{cat}}^{-1}$ assuming cathode loadings of 3 and 1 $\text{mg}_{\text{cat}} \text{cm}^{-2}$, respectively. The underlying reason behind our choice of small currents lies in the limitations of both the SFC-ICP-MS and EC-MS setups to effectively deplete gas bubbles under high turnover currents, which lead to loss of potential control upon blockage of the SFC V-shaped channel. Therefore, such high mass-normalized current densities can only be achieved in other setups. Following up the suggestions by reviewers #2 and #3, we have performed long-term stability tests on both a-MoS_{3-x} and c-MoS₂ electrodeposited thin films on an H-cell setup for 5 hours under 1) start-up/shutdown between -100 mA cm^{-2} and 0 V vs. RHE and 2) constant current densities of -100 mA cm^{-2} . The high hydrogen evolution rates (loading-normalized currents of -741 mA $\text{mg}_{\text{cat}}^{-1}$) are within the range anticipated for non-noble PEMWE cathodes under realistic operating conditions. Interestingly, the stability of a-MoS_{3-x} is observed here to be 10-fold higher than that of c-MoS₂, which we believe it arises from a change in the intrinsic c-MoS₂ site stability at higher HER rates. The results are discussed in a dedicated section of the revised manuscript as well as in the discussion and experimental, included below:

Study of long-term stability: H-cell measurements

In order to validate the observed trends at application-relevant operating currents, long-term stability measurements were performed at -100 mA cm^{-2} (ca. -741 mA $\text{mg}_{\text{cat}}^{-1}$) in an H-cell configuration for both freshly-prepared c-MoS₂ and a-MoS_{3-x} (see Figures S23-S24 for results and S25 for physicochemical characterization). To assess the impact of intermittent operation, H-cell measurements were performed for 5 hours under constant HER load as well as under start-up/shutdown conditions (start-stop cycles from -100 mA $\text{cm}^{-2}_{\text{geom}}$ to 0 VRHE, 2 mins per pulse). It was observed that only Mo dissolution could be quantitatively assessed, which leads us to conclude that under high HER reaction rates S loss predominantly takes place by gaseous H₂S evolution for both catalysts, as S does not remain in the liquid phase. The resulting S-

numbers obtained after Mo quantification of H-cell liquid aliquots (Figure 8) showcase that constant operation yields 3 to 5 times higher stability for both catalysts (1.2 to 2 times higher loading-normalized dissolution, see Figure S24 and Table S3 for values), and that the stability trends are inversed compared with those at low current densities: *a*-MoS_{3-x} stability is almost 10-fold higher than *c*-MoS₂, regardless of the operation mode. Post-mortem, *ex-situ* SEM imaging demonstrated that catalyst degradation by mechanical delamination was not present in either *c*-MoS₂ (see A-C Figure S26) or *a*-MoS_{3-x} (see A-C Figure S28), therefore not being responsible for the observed trends. Analogous conclusions were drawn from XRD measurements, whereby all diffractograms remained unchanged after testing (see D-F Figures S26 and S28). In contrast, XPS and Raman measurements showcased a clear loss of the distinct E_{12g} and A_{1g} Raman modes in *c*-MoS₂ (see G-I Figure S26) as well as an increased presence of oxidized Mo⁵⁺O_xS_y:Mo⁶⁺ species at the expense of Mo⁴⁺ from MoS₂ (ca. 9:42 at. % after intermittent operation, see Figure S27). Interestingly, post-mortem *a*-MoS_{3-x} presented almost negligible Mo⁵⁺O_xS_y:Mo⁶⁺ surface contents (ca. 2:1 at. %, see Figure S29) which, along with the loss of S₂²⁻_{bridg}/S₂²⁻_{term}/S₂²⁻_{ap} Raman modes (ca. 550, 520 and 450 cm⁻¹) and the well-reported appearance of both a band at 430 cm⁻¹ (resembling A_{1g}) and MoO_x-related Raman features upon environment exposure (broad band at 800-1000 cm⁻¹), corroborate the structural transformation to the MoS_{2-x} structure via S loss and undercoordinated Mo site generation. Therefore, the initially high *c*-MoS₂ stability observed worsens at high HER reaction rates in contrast with *a*-MoS_{3-x}, which presents similar Mo S-numbers at low and high HER rates (~10⁵-10⁶, see Figure 5, 8, S19 and S21). We must note, however, that Mo S-numbers might be overestimated given the S loss via volatile H₂S.

Figure 8. Compilation of Mo S-numbers obtained during long-term H-cell measurements consisting of constant HER operation (5 h, -100 mA cm⁻²; left column) and start-up/shutdown HER intermittent operation alternating (5 h, -100 mA cm⁻² to 0 V vs. RHE; right column).

Figure S23. Compilation of potential vs. time curves obtained for a-MoS_{3-x} (light blue) and c-MoS₂ (dark blue) during H-cell measurements consisting of 5 hours constant galvanostatic hold at -100 mA cm⁻² (top panel) and 5 hours undergoing start-up/shutdown HER holds alternating from -100 mA cm⁻² to 0 V vs. RHE (bottom panel).

Figure S24. Compilation of total (left) and loading-normalized (right) integrated dissolution of Mo for $a\text{-MoS}_{3-x}$ (light blue) and $c\text{-MoS}_2$ (dark blue) obtained after H-cell measurements consisting of 5 hours constant galvanostatic hold at -100 mA cm^{-2} (solid columns) and after 5 hours undergoing start-up/shutdown HER holds alternating from -100 mA cm^{-2} to 0 V vs. RHE (dashed columns).

Excerpts from the Discussion section:

‘Second, all $[\text{Mo}_3\text{S}_{13}]$ -based catalysts present 100-fold lower stability under HER potentials than $c\text{-MoS}_2$ at low HER reaction rates, based on the S-number metric: for Mo: $S \sim 10^5$ vs. 10^7 ; for S: $S \sim 10^4$ vs. 10^6 . At high reaction rates, however, $a\text{-MoS}_{3-x}$ stability is almost 10-fold higher than $c\text{-MoS}_2$ ($S \sim 10^6$ vs. 10^5 under constant HER hold).’

‘However, the different Mo S-numbers obtained for $c\text{-MoS}_2$ compared to $a\text{-MoS}_{3-x}$ at low and high HER rates (100-fold higher and 10-fold lower, see Figures 5 and 8) would contradict this hypothesis: either Mo^{3+} hydride sites would be (de)stabilized (i.e. different intrinsic activity) or an alternative HER pathway would be involved.’

‘At high HER rates, given that $a\text{-MoS}_{3-x}$ is 10-fold more stable (Figure 8), either 1) $a\text{-MoS}_{3-x}$ should present 100-fold lower intrinsic stability at equivalent HER rates to compensate for the 10-fold higher TOFs in $c\text{-MoS}_2$ or 2) $c\text{-MoS}_2$ should have a 100-fold higher density of Mo-H sites, the latter being incorrect as mentioned earlier.’

'Indeed, c-MoS₂ and [Mo₃S₁₃]-based catalysts presented, at low HER rates, consistently different stabilities across different HER rates to those of c-MoS₂, regardless of the operating conditions (Figures S20-S22). However, the stability crossover found when comparing low and high HER rates for c-MoS₂ (Mo S-number ~10⁷ vs ~10⁵, see Figure 5 and 8) suggests that intrinsic active site stabilities are not constant, and are highly dependent on the preferential HER and dissolution pathway. Such hypothesis is reinforced when analysing the post-mortem S-to-Mo XPS ratios after long-term testing, which indicate that near-surface sulphur species are favourably retained in c-MoS₂ when compared to a-MoS_{3-x} (S-to-Mo ca. 1.5 vs. 0.4, see Table S4). This leads us to believe that, for c-MoS₂, Mo-H formation cannot solely be responsible for the HER activity. At low HER rates, given the experimental evidence of low Mo dissolution and almost negligible S loss (Figures 5, 7 and S17-S19), we hypothesize that unsaturated S²⁻ sites are primarily involved as proton-accepting groups. At higher HER rates (Figure 8), given the almost stoichiometric near-surface S-to-Mo ratios found in c-MoS₂ after long-term testing (ca. 1.5, see Table S4), we propose that Mo³⁺ hydride formation (steps I' and II' in Scheme 2) would be responsible for the HER activity as well as the dissolution pathways via stoichiometric loss of Mo and S in step VI' in Scheme 2.'

'In contrast, the stability of [Mo₃S₁₃]-based catalysts markedly increased up to S-numbers ~10⁵ and ~10⁶ under higher current densities of -10 mA cm⁻² (Figure S20-22 ESI) and -100 mA cm⁻² (Figure 8), respectively. Hence, MoS_x-based PEMWEs would initially seem bound to operate under low constant current loads when using c-MoS₂, while high-current operation even under intermittent mode would be suited for [Mo₃S₁₃]-based catalysts. It is noteworthy to mention that lifetimes of IrO_x and non-noble anode catalysts were dramatically extended in membrane electrode assembly (MEA) environments^{72, 73}, which could also apply to a [Mo₃S₁₃]-based cathode even more than that observed after long-term H-cell measurements.'

H-Cell Stability Measurements

Electrochemical long-term stability tests were conducted in commercial H-cells (Pine Research). Each compartment was filled with 30 mL of 0.1 M HClO₄, prepared from 70% HClO₄ (Suprapur, Merck). Prior to measurement, the electrolyte was saturated with argon for 30 minutes and maintained above the solution throughout the experiment. The setup included a working electrode of electrodeposited MoS_x on glassy carbon (SIGRADUR G, HTW) and an Ag/AgCl reference electrode (Metrohm, Germany) in one compartment. A glassy carbon counter electrode (SIGRADUR G, HTW) in the opposite compartment minimized Mo ion redeposition. A magnetic stirrer at 800 RPM ensured consistent convection, preventing local

concentration gradients. Control of potential and current was achieved using a VSP-300 potentiostat (Biologic). The study employed two distinct protocols: the first involved a consistent -100 mA cm^{-2} galvanostatic hold for 5 hours to evaluate stability. The second protocol consisted of alternating -100 mA cm^{-2} pulses with $0 V_{RHE}$, each for 2 minutes, continuing for a total duration of 5 hours. In addition, a 30-minute $0 V_{RHE}$ potentiostatic hold was applied before and after each protocol. Ohmic resistance was determined by impedance spectroscopy, specifically at the high-frequency interception of the $Re(Z)$ axis, to fully compensate for iR drop during the protocols. Sample aliquots ($V = 500 \mu\text{L}$) were taken at predetermined intervals, and analyzed offline using ICP-MS. Consistency in the total volume of electrolyte, maintained at 30 mL in both compartments, was ensured throughout the experiment.

4- It would have been interesting to quantify the amount of S and Mo dissolved with respect to their initial loading on the electrode (as it has been done in the recent paper of Zavala and al. *ACS Catal.* 2023, 13, 1221–1229).

We thank the reviewer for the suggestion. We amended this in the revised version of the manuscript and Supporting information by providing the relative dissolution amounts of Mo and S with respect to the initial Mo and S catalyst loading. For ease of convenience, there resulting figures are included as follows:

Figure S2. Graphical representation of the loading-normalized integrated dissolution of Mo (top, blue) and S (bottom, yellow) as a function of the LPL. For electrochemical protocol, see Figure 1. Scan rate: 5 mV s^{-1} .

Figure S5. Loading-normalized integrated dissolution of Mo (first panel, blue) and S (second panel, yellow) and the corresponding experimental dissolution onset potentials of Mo (third panel) and S (fourth panel) as a function of the LPL. CVs recorded from 0 V_{RHE} to LPLs in the range $-0.1 \leq E_{UPL} \leq -0.5 \text{ V}$. Scan rate: 5 mV s^{-1} .

Figure S7. Loading-normalized dissolution data for Mo (top) and S (bottom) integrated at each consecutive $-1 \text{ mA cm}^{-2}_{\text{geom}}$ HER holds (left, “start-up” cycle) and 0 V_{RHE} (right, “shutdown” cycle). Labels: $a\text{-MoS}_{3-x}$ (light blue/yellow) and $c\text{-MoS}_2$ (dark blue/yellow).

Figure S10. Total (first panel) and loading-normalized (second panel) dissolution data for Mo (left) and S (right) integrated at each consecutive $-1 \text{ mA cm}^{-2}_{\text{geom}}$ HER holds. Compilation of S-numbers obtained for a-MoS_{3-x}, [Mo₃S₁₃]²⁻ and [Mo₃S₁₃]-N-CNTs during start-up/shut-down stress tests for Mo and S under HER conditions (third panel). Fourth panel: loading-normalized Mo dissolution and corresponding S-numbers(e⁻) during 0V vs. RHE holds.

Figure S11. Total (top panel) and loading-normalized (middle panel) integrated dissolution of Mo (blue) and S (yellow) and the corresponding experimental dissolution onset potentials (bottom panel) for a-MoS_{3-x} and c-MoS₂ and as a function of the UPL. Scan rate: 5 mV s⁻¹.

Figure S13. Total (top panel) and loading-normalized (middle panel) integrated dissolution of Mo (blue) and S (yellow) and the corresponding experimental dissolution onset potentials (bottom panel) for the [Mo₃S₁₃]-based catalysts (a-MoS_{3-x}, [Mo₃S₁₃]²⁻ and MoS_x-N-CNT) as a function of the UPL. Scan rate: 5 mV s⁻¹.

Figure S19. Compilation of loading-normalized integrated dissolution of Mo (top) and S (bottom) for c-MoS₂ and a-MoS_{3-x} during sequential galvanostatic holds under HER conditions (left) and during 0V vs. RHE holds (right).

Figure S22. Compilation of loading-normalized integrated dissolution of Mo (top) and S (bottom) obtained for a-MoS_{3-x}, [Mo₃S₁₃]²⁻ and [Mo₃S₁₃]-N-CNTs during sequential galvanostatic holds under HER conditions (left) and during 0V vs. RHE holds (right).

Figure S24. Compilation of total (left) and loading-normalized (right) integrated dissolution of Mo for a-MoS_{3-x} (light blue) and c-MoS_2 (dark blue) obtained after H-cell measurements consisting of 5 hours constant galvanostatic hold at -100 mA cm^{-2} (solid columns) and after 5 hours undergoing start-up/shutdown HER holds alternating from -100 mA cm^{-2} to 0 V vs. RHE (dashed columns).

Reviewer #3 (Remarks to the Author):

Since MoS₂ is regarded as a competitive to replace platinum in PEMWEs, the authors investigated the stability of MoS₂ cathodes with different crystal structures (a-MoS₂ and c-MoS₂) under acidic conditions. This is a pretty new perspective in researching the over-investigated electrocatalysts and they found that a-MoS₂ possessed a high activity while c-MoS₂ had better stability at the HER operating potential. Furthermore, the cathodic dissolution pathways and HER mechanism were detailed discussed. They concluded that c-MoS₂ presented the best trade-off between activity and stability if PEMWEs were to operate under constant load. However, the most doubtful points are the not excellent activity of c-MoS₂ according to the LSV data and not realistic working condition (up to 10 mA/cm²). To the best of our knowledge, many catalysts with high activity and high stability under acidic conditions have been designed [Sci. Adv. 9, eadh2285 (2023)], [Nat. Commun. 13, 7784 (2022)], which are even much better than a-MoS₂. Thus, I would like to recommend a major revision and improvement before further consideration of this manuscript. Here are the detailed comments:

We thank the reviewer for the time dedicated to review our manuscript, in addition to the insightful comments on our work. We will aim at addressing the reviewer's main concerns point by point in the following section.

1. This work only studied the ion dissolution of the catalyst under the small current density (up to 10 mA cm⁻²). While a high current density, at least larger than 100 mA cm⁻², is required to meet the practical operating condition. Under such a high current density, the failure mechanism of the catalyst may change. Therefore, I think the stability at high current densities is also noteworthy to be investigated.

We appreciate the comment from the reviewer concerning the transferability of our results to PEMWE-relevant operating conditions. As pointed out in our earlier response to reviewer #2 (see discussion above), the choice of such currents lies in the limitations of both the SFC-ICP-MS and EC-MS setups: gas build-up at the confined reaction compartments leads to loss of potential control, prevents us from exerting high current densities. The low HER activities observed by both a-MoS_{3-x} and c-MoS₂ arise from the very low loadings present in our catalysts (ranging from 60-100 $\mu\text{g}_{\text{cat}} \text{cm}^{-2}$) which are below the ones generally reported non-noble metal electrocatalysts, targeted at maximizing mass-normalized current densities.

Following up the suggestions by reviewers #2 and #3, we have performed long-term stability tests on both a-MoS_{3-x} and c-MoS₂ electrodeposited thin films on an H-cell setup for 5 hours under 1) start-up/shutdown between -100 mA cm⁻² and 0 V vs. RHE and 2) constant current densities of -100 mA cm⁻². The high hydrogen evolution rates (loading-normalized currents of

-741 mA mg_{cat}⁻¹) are within the range anticipated for non-noble PEMWE cathodes under realistic operating conditions, as these range between 333 and 1000 mA mg_{cat}⁻¹ under PEMWE operation at 1 A cm⁻², assuming cathode loadings of 3 and 1 mg_{cat} cm⁻², respectively. Interestingly, the stability of a-MoS_{3-x} is observed here to be 10-fold higher than that of c-MoS₂, which we believe they arise from a change in the intrinsic site stability at higher HER rates. The results are discussed in a dedicated section of the revised manuscript as well as in the discussion, included below:

Study of long-term stability: H-cell measurements

In order to validate the observed trends at application-relevant operating currents, long-term stability measurements were performed at -100 mA cm⁻² (ca. -741 mA mg_{cat}⁻¹) in an H-cell configuration for both freshly-prepared c-MoS₂ and a-MoS_{3-x} (see Figures S23-S24 for results and S25 for physicochemical characterization). To assess the impact of intermittent operation, H-cell measurements were performed for 5 hours under constant HER load as well as under start-up/shutdown conditions (start-stop cycles from -100 mA cm⁻²_{geom} to 0 V_{RHE}, 2 mins per pulse). It was observed that only Mo dissolution could be quantitatively assessed, which leads us to conclude that under high HER reaction rates S loss predominantly takes place by gaseous H₂S evolution for both catalysts, as S does not remain in the liquid phase. The resulting S-numbers obtained after Mo quantification of H-cell liquid aliquots (Figure 8) showcase that constant operation yields 3 to 5 times higher stability for both catalysts (1.2 to 2 times higher loading-normalized dissolution, see Figure S24 and Table S3 for values), and that the stability trends are inversed compared with those at low current densities: a-MoS_{3-x} stability is almost 10-fold higher than c-MoS₂, regardless of the operation mode. Post-mortem, ex-situ SEM imaging demonstrated that catalyst degradation by mechanical delamination was not present in either c-MoS₂ (see A-C Figure S26) or a-MoS_{3-x} (see A-C Figure S28), therefore not being responsible for the observed trends. Analogous conclusions were drawn from XRD measurements, whereby all diffractograms remained unchanged after testing (see D-F Figures S26 and S28). In contrast, XPS and Raman measurements showcased a clear loss of the distinct E¹_{2g} and A_{1g} Raman modes in c-MoS₂ (see G-I Figure S26) as well as an increased presence of oxidized Mo⁵⁺O_xS_y:Mo⁶⁺ species at the expense of Mo⁴⁺ from MoS₂ (ca. 9:42 at. % after intermittent operation, see Figure S27). Interestingly, post-mortem a-MoS_{3-x} presented almost negligible Mo⁵⁺O_xS_y:Mo⁶⁺ surface contents (ca. 2:1 at. %, see Figure S29) which, along with the loss of S₂²⁻_{bridg}/S₂²⁻_{term}/S₂²⁻_{ap} Raman modes (ca. 550, 520 and 450 cm⁻¹) and the well-reported appearance of both a band at 430 cm⁻¹ (resembling A_{1g}) and MoO_x-related Raman features upon environment exposure (broad band at 800-1000 cm⁻¹), corroborate the structural

transformation to the MoS_{2-x} structure via S loss and undercoordinated Mo site generation. Therefore, the initially high $c\text{-MoS}_2$ stability observed worsens at high HER reaction rates in contrast with $a\text{-MoS}_{3-x}$, which presents similar Mo S-numbers at low and high HER rates ($\sim 10^5 - 10^6$, see Figure 5, 8, S19 and S21). We must note, however, that Mo S-numbers might be overestimated given the S loss via volatile H_2S .

Figure 8. Compilation of Mo S-numbers obtained during long-term H-cell measurements consisting of constant HER operation (5 h, -100 mA cm^{-2} ; left column) and start-up/shutdown HER intermittent operation alternating (5 h, -100 mA cm^{-2} to 0 V vs. RHE; right column).

Figure S23. Compilation of potential vs. time curves obtained for $a\text{-MoS}_{3-x}$ (light blue) and $c\text{-MoS}_2$ (dark blue) during H-cell measurements consisting of 5 hours constant galvanostatic hold at -100 mA cm^{-2} (top panel) and 5 hours undergoing start-up/shutdown HER holds alternating from -100 mA cm^{-2} to 0 V vs. RHE (bottom panel).

Figure S24. Compilation of total (left) and loading-normalized (right) integrated dissolution of Mo for $a\text{-MoS}_{3-x}$ (light blue) and $c\text{-MoS}_2$ (dark blue) obtained after H-cell measurements consisting of 5 hours constant galvanostatic hold at -100 mA cm^{-2} (solid columns) and after 5 hours undergoing start-up/shutdown HER holds alternating from -100 mA cm^{-2} to 0 V vs. RHE (dashed columns).

Excerpts from the Discussion section:

‘Second, all $[\text{Mo}_3\text{S}_{13}]$ -based catalysts present 100-fold lower stability under HER potentials than $c\text{-MoS}_2$ at low HER reaction rates, based on the S-number metric: for Mo: $S \sim 10^5$ vs. 10^7 ; for S: $S \sim 10^4$ vs. 10^6 . At high reaction rates, however, $a\text{-MoS}_{3-x}$ stability is almost 10-fold higher than $c\text{-MoS}_2$ ($S \sim 10^6$ vs. 10^5 under constant HER hold).’

‘However, the different Mo S-numbers obtained for $c\text{-MoS}_2$ compared to $a\text{-MoS}_{3-x}$ at low and high HER rates (100-fold higher and 10-fold lower, see Figures 5 and 8) would contradict this hypothesis: either Mo^{3+} hydride sites would be (de)stabilized (i.e. different intrinsic activity) or an alternative HER pathway would be involved.’

‘At high HER rates, given that $a\text{-MoS}_{3-x}$ is 10-fold more stable (Figure 8), either 1) $a\text{-MoS}_{3-x}$ should present 100-fold lower intrinsic stability at equivalent HER rates to compensate for the 10-fold higher TOFs in $c\text{-MoS}_2$ or 2) $c\text{-MoS}_2$ should have a 100-fold higher density of Mo-H sites, the latter being incorrect as mentioned earlier.’

‘Indeed, *c*-MoS₂ and [Mo₃S₁₃]-based catalysts presented, at low HER rates, consistently different stabilities across different HER rates to those of *c*-MoS₂, regardless of the operating conditions (Figures S20-S22). However, the stability crossover found when comparing low and high HER rates for *c*-MoS₂ (Mo S-number ~10⁷ vs ~10⁵, see Figure 5 and 8) suggests that intrinsic active site stabilities are not constant, and are highly dependent on the preferential HER and dissolution pathway. Such hypothesis is reinforced when analysing the post-mortem S-to-Mo XPS ratios after long-term testing, which indicate that near-surface sulphur species are favourably retained in *c*-MoS₂ when compared to *a*-MoS_{3-x} (S-to-Mo ca. 1.5 vs. 0.4, see Table S4). This leads us to believe that, for *c*-MoS₂, Mo-H formation cannot solely be responsible for the HER activity. At low HER rates, given the experimental evidence of low Mo dissolution and almost negligible S loss (Figures 5, 7 and S17-S19), we hypothesize that unsaturated S²⁻ sites are primarily involved as proton-accepting groups. At higher HER rates (Figure 8), given the almost stoichiometric near-surface S-to-Mo ratios found in *c*-MoS₂ after long-term testing (ca. 1.5, see Table S4), we propose that Mo³⁺ hydride formation (steps I’ and II’ in Scheme 2) would be responsible for the HER activity as well as the dissolution pathways via stoichiometric loss of Mo and S in step VI’ in Scheme 2. ‘

‘In contrast, the stability of [Mo₃S₁₃]-based catalysts markedly increased up to S-numbers ~10⁵ and ~10⁶ under higher current densities of -10 mA cm⁻² (Figure S20-22 ESI) and -100 mA cm⁻² (Figure 8), respectively. Hence, MoS_x-based PEMWEs would initially seem bound to operate under low constant current loads when using *c*-MoS₂, while high-current operation even under intermittent mode would be suited for [Mo₃S₁₃]-based catalysts. It is noteworthy to mention that lifetimes of IrO_x and non-noble anode catalysts were dramatically extended in membrane electrode assembly (MEA) environments^{72, 73}, which could also apply to a [Mo₃S₁₃]-based cathode even more than that observed after long-term H-cell measurements.’

H-Cell Stability Measurements

Electrochemical long-term stability tests were conducted in commercial H-cells (Pine Research). Each compartment was filled with 30 mL of 0.1 M HClO₄, prepared from 70% HClO₄ (Suprapur, Merck). Prior to measurement, the electrolyte was saturated with argon for 30 minutes and maintained above the solution throughout the experiment. The setup included a working electrode of electrodeposited MoS_x on glassy carbon (SIGRADUR G, HTW) and an Ag/AgCl reference electrode (Metrohm, Germany) in one compartment. A glassy carbon counter electrode (SIGRADUR G, HTW) in the opposite compartment minimized Mo ion redeposition. A magnetic stirrer at 800 RPM ensured consistent convection, preventing local concentration gradients. Control of potential and current was achieved using a VSP-300

potentiostat (Biologic). The study employed two distinct protocols: the first involved a consistent -100 mA cm^{-2} galvanostatic hold for 5 hours to evaluate stability. The second protocol consisted of alternating -100 mA cm^{-2} pulses with $0 V_{RHE}$, each for 2 minutes, continuing for a total duration of 5 hours. In addition, a 30-minute $0 V_{RHE}$ potentiostatic hold was applied before and after each protocol. Ohmic resistance was determined by impedance spectroscopy, specifically at the high-frequency interception of the $Re(Z)$ axis, to fully compensate for iR drop during the protocols. Sample aliquots ($V = 500 \mu\text{L}$) were taken at predetermined intervals, and analyzed offline using ICP-MS. Consistency in the total volume of electrolyte, maintained at 30 mL in both compartments, was ensured throughout the experiment.

2. The long-term stability of these two electrodes needs to be supplemented to quantify the effect of sulfur dissolution on the electrode stability.

Following up with the previous point raised by the reviewer, we monitored Mo and S dissolution during H-cell measurements under constant and intermittent HER operation for both a-MoS_{3-x} and c-MoS₂ electrodeposited thin films by extracting aliquots from the working electrode compartment. Although we could clearly track Mo dissolution over the course of the long-term experiments, S dissolution could not be reliably monitored. We believe that this due two main factors. First, the ca. 3-4 orders of magnitude larger electrolyte volume required for H-cell measurements which decreased sensitivity toward ICP-MS quantification due to lower effective dissolved Mo/S concentrations. Second, and perhaps most important, the preferential loss of S under HER conditions via gaseous H₂S, as pointed out in our EC-MS measurements compiled in Figure 7 in the revised manuscript.

As an indirect approach to assess S loss in both materials, we analysed the S-to-Mo ratios of pristine and long-term tested a-MoS_{3-x} and c-MoS₂ with XPS (see Figures S25-S29 in Supporting Information), its results being compiled in Table S3 in the Supporting information. For both a-MoS_{3-x} and c-MoS₂ electrodeposited thin films the S dissolution was not the critical factor when describing their long-term stability. It was found, particularly for c-MoS₂, that Mo dissolution played a much major role in the S-number metrics considering the relatively low S loss at the near surface compared to the ca. 10% loss of Mo versus its initial loading. Interestingly, a-MoS_{3-x} presented a 10-fold higher stability while S was preferentially lost at the near-surface based on the S-to-Mo ratios observed.

3. More characterizations, such as XRD, are required to confirm that electrodes with two different crystal structures have been successfully prepared.

We thank the reviewer for the suggestion. Following up with our earlier response to reviewer #1, these materials do not present long range periodical order in neither their pristine nor their tested state. This can be clearly observed when looking at the XRD patterns of both a-MoS_{3-x} and c-MoS₂ (see Figure S25 Supporting Information and R1 from response to Reviewer #1).

However, these materials present distinctive Raman modes highly dependent on the structural motifs present (see *ACS Catal.* 2016, 6, 7790-7798) which allow to clearly distinguish the $[\text{Mo}_3\text{S}_{13}]^{2-}$ -based catalysts from the *c*- MoS_2 /crystalline MoS_2 counterparts, as confirmed by the ex-situ Raman spectra obtained for pristine and tested *a*- MoS_{3-x} and *c*- MoS_2 (see E-F Figure S25, and G-I Figure S26 and S28 in the Supporting information). In addition, it is well reported in the literature that the S 2p XPS region is a footprint for both anodically and cathodically electrodeposited MoS_x materials (see *ACS Catal.* 2013, 3, 2002-2011), arising from differences in the relative abundance of S_2^{2-} and S^{2-} moieties in their structure, clearly shown in our measurements. To provide a more in-depth view of the amorphous MoS_x structures obtained, the revised version of the Supporting Information includes both XRD pattern and Raman spectra for the pristine materials (see Figure S25), these being briefly discussed in the revised manuscript and appended in the response to the second question of reviewer #1.

4. In addition to EC-MS and XPS, SEM and other characterizations needs to be conducted after the stability test to rule out other factors besides dissolution that cause electrode failure.

We appreciate the concerns from the reviewer regarding the long-term electrode failure. After conducting additional post-mortem measurements on the tested electrodes employing SEM (A-B Figure S25, A-C Figure S26 and A-C Figure S28), we observed that the samples presented no delamination or mechanical failure. Therefore, we believe that the integrity of the electrode is not compromised after long-term HER testing at high current densities. We refer to the reviewer to the answer provided to second question of reviewer #1 for the full physicochemical characterization and detailed description of both Raman and XPS measurements performed.

5. Sulfur dissolution of c-MoS2 under “start-up/shutdown” HER holds (Figure 4) demonstrates that a large amount of sulfur dissolution occurs in the first “start-up/shutdown” cycle (around 1600 s) while remaining at a low value in subsequent cycles. Please explain this phenomenon.

We thank the reviewer for his comment. Cathodically-electrodeposited MoS_2 (*c*- MoS_2) is known to have a crystalline structure similar to that of crystalline 2H- MoS_2 . It has been previously reported that 2H- MoS_2 degradation arises from the high driving force for its decomposition close to open circuit potential initially observed by Ledendecker et al. using online ICP-MS (*Angew Chem Int Ed* 2017, **56**, 9767-9771) and later corroborated with DFT calculations by Chorkendorff/Norskov's groups (*ACS Energy Lett.* 2021, 6, 6, 2268–2274). This is once more shown to be the case at start-up/shutdown intermittent operation at low HER rates as shown in Figures 4 and 5 of the main manuscript even though resting potentials were still away from OCP. We believe that the significantly low S dissolution at low HER rates and subsequent fast stabilization stems from the HER mechanism and the Mo dissolution pathways. Although a Mo^{3+} hydride has been proposed as the universal active site in Mo-based HER catalysts, we believe that unsaturated sulphur sites primarily act as proton-accepting groups in *c*- MoS_2 at low HER rates, yielding almost no net sulphur loss under HER turnovers (see Figure 7 for EC-MS measurements). On the other hand, undercoordinated Mo^{3+} would be the intermediate responsible for Mo loss once the potential is reversed to close to 0 V vs. RHE and at much higher HER rates where many more S vacancies can be electrochemically formed to

yield undercoordinated Mo^{3+} sites (see *Nat Commun* 2017, **8**, 15113), yielding HER and dissolution pathways following those of Scheme 2. The correlation between S loss and HER/dissolution pathways is discussed in the revised version of the manuscript as follows:

Indeed, c-MoS₂ and [Mo₃S₁₃]-based catalysts presented, at low HER rates, consistently different stabilities across different HER rates to those of c-MoS₂, regardless of the operating conditions (Figures S20-S22). However, the stability crossover found when comparing low and high HER rates for c-MoS₂ (Mo S-number $\sim 10^7$ vs $\sim 10^5$, see Figure 5 and 8) suggests that intrinsic active site stabilities are not constant, and are highly dependent on the preferential HER and dissolution pathway. Such hypothesis is reinforced when analysing the post-mortem S-to-Mo XPS ratios after long-term testing, which indicate that near-surface sulphur species are favourably retained in c-MoS₂ when compared to a-MoS_{3-x} (S-to-Mo ca. 1.5 vs. 0.4, see Table S4). This leads us to believe that, for c-MoS₂, Mo-H formation cannot solely be responsible for the HER activity. At low HER rates, given the experimental evidence of low Mo dissolution and almost negligible S loss (Figures 5, 7 and S17-S19), we hypothesize that unsaturated S²⁻ sites are primarily involved as proton-accepting groups. At higher HER rates (Figure 8), given the almost stoichiometric near-surface S-to-Mo ratios found in c-MoS₂ after long-term testing (ca. 1.5, see Table S4), we propose that Mo³⁺ hydride formation (steps I' and II' in Scheme 2) would be responsible for the HER activity as well as the dissolution pathways via stoichiometric loss of Mo and S in step VI' in Scheme 2.

6. Some formatting mistakes should be corrected, such as “a-MoS_{3-x} and c-MoS₂” in line 82. Please double-check the manuscript.

We thank the reviewer for pointing out the formatting inconsistencies. The revised manuscript has been thoroughly proof-read and all formatting issues have been amended, but we believe that he/she is referring to the actual terminology employed to refer to anodically-electrodeposited MoS_x and cathodically-electrodeposited MoS_x. Across the manuscript we refer to each of these as a-MoS_{3-x} and c-MoS₂ (a- stemming from anodic and c- stemming from cathodic), respectively. Given that the S-to-Mo stoichiometry in anodically-electrodeposited MoS_x is close to 3 after electrolyte immersion in acid (see Table 1 *ACS Catal.* 2013, 3, 2002–2011), we opted to refer to it as a-MoS_{3-x}, in line with other reports in the literature. We hope that this clears out any potential misunderstanding regarding the chosen terminology.

REVIEWER COMMENTS

Reviewer #1 (Remarks to the Author):

The authors have addressed the reviewers' previous questions in a thorough and detailed manner. The reviewer does not have any further questions.

Reviewer #2 (Remarks to the Author):

The authors have taken into account all my comments.
The paper can be published in its present shape.

Reviewer #3 (Remarks to the Author):

The authors have provided comprehensive and detailed responses to the reviewers' comments, especially to those from my side. I have also observed a considerable improvement in the manuscript through necessary revisions.

However, I still have doubts about the discussion on the long-term stability of the two catalysts under high current density. According to the stability test, it seems that a-MoS_{3-x} has better stability due to its less performance degradation after operation for 5 hours (Figure S23). However, the structure of a-MoS_{3-x} is unstable because it leaches more Mo (Figure S24). The results of these two experiments appear to be contradictory, so how did the authors conclude that c-MoS₂ has better stability than a-MoS_{3-x}?

Response to reviewers

Reviewer #1 (Remarks to the Author):

The authors have addressed the reviewers' previous questions in a thorough and detailed manner. The reviewer does not have any further questions.

We would like to thank the reviewer for the positive feedback on the revisions made, which we believe strengthened the conclusions of the work presented.

Reviewer #2 (Remarks to the Author):

*The authors have taken into account all my comments.
The paper can be published in its present shape.*

We would like to thank once more reviewer #2 for the insightful comments, and we greatly appreciate that after implementing these in the manuscript he/she deemed our work to be published in its current state.

Reviewer #3 (Remarks to the Author):

The authors have provided comprehensive and detailed responses to the reviewers' comments, especially to those from my side. I have also observed a considerable improvement in the manuscript through necessary revisions. However, I still have doubts about the discussion on the long-term stability of the two catalysts under high current density. According to the stability test, it seems that a-MoS_{3-x} has better stability due to its less performance degradation after operation for 5 hours (Figure S23). However, the structure of a-MoS_{3-x} is unstable because it leaches more Mo (Figure S24). The results of these two experiments appear to be contradictory, so how did the authors conclude that c-MoS₂ has better stability than a-MoS_{3-x}?

We would like to thank the reviewer for acknowledging the efforts made to tackle all reviewers' comments as comprehensively and detailed as possible.

With regards to the reviewer's additional comments, we must point out that during the long-term HER measurements it was indeed c-MoS₂ (dark blue in Figure S24) the catalyst which dissolved almost ten times more than a-MoS_{3-x} (light blue in Figure S24). This was additionally displayed in Table S3, where the final loading-normalized Mo loss and Mo-S values were compiled for a more straightforward comparison. For ease of convenience both Figure S24 and Table S3 have been included in the response. Therefore, at high current densities a-MoS_{3-x} is indeed more stable than c-MoS₂ not only in terms of electrochemical performance (as shown in Figure S23 and pointed out by reviewer #3) but also in terms of stability, showcased by the almost 10-fold higher S-numbers obtained for a-MoS_{3-x} (see Figure 8 main manuscript, also included in the response). The presented Mo S-numbers in Figure 8 were calculated using the quantified Mo dissolution during the long-term H-cell measurement, shown in Figure S24.

The 10-fold higher a-MoS_{3-x} stability compared to c-MoS₂ at high current densities was stated and discussed, as shown in the excerpts below from the section ‘Study of long-term stability: H-cell measurements’:

*‘The resulting S-numbers obtained after Mo quantification of H-cell liquid aliquots (Figure 8) showcase that constant operation yields 3 to 5 times higher stability for both catalysts (1.2 to 2 times higher loading-normalized dissolution, see Figure S24 and Table S3 for values), and that the stability trends are inversed compared with those at low current densities: **a-MoS_{3-x} stability is almost 10-fold higher than c-MoS₂, regardless of the operation mode.**’*

We must point out that the a-MoS_{3-x} initial structure is not retained under hydrogen turnovers, as it undergoes a structural transformation to yield undercoordinated Mo sites. This was discussed in the same section of the manuscript (see also Figure S29 for Raman/XPS data), as well as in the HER/dissolution pathway shown in Scheme 2:

‘Interestingly, post-mortem a-MoS_{3-x} presented almost negligible Mo⁵⁺O_xS_y:Mo⁶⁺ surface contents (ca. 2:1 at. %, see Figure S29) which, along with the loss of S₂²⁻_{bridg}/S₂²⁻_{term}/S₂²⁻_{ap} Raman modes (ca. 550, 520 and 450 cm⁻¹) and the well-reported appearance of both a band at 430 cm⁻¹ (resembling A_{1g}) and MoO_x-related Raman features upon environment exposure (broad band at 800-1000 cm⁻¹), corroborate the structural transformation to the MoS_{2-x} structure via S loss and undercoordinated Mo site generation.’

The long-term stability results yielded opposed results to those shown for lower current densities with the SFC-ICP-MS setup: at low current ranges (-1 to -10 mA cm⁻²) c-MoS₂ is more stable than a-MoS_{3-x} (see Mo S-numbers in Figure 5 and Figure 8). We ascribed this apparent contradiction (crossover in stability trends when comparing high and low current densities) to a stark change in the inherent active site stability in c-MoS₂. At low HER currents, unsaturated S²⁻ sites present in c-MoS₂ are the predominant active sites, and therefore Mo (see Figure 4) and S dissolution (see Figure 7) are negligible unless 1) undercoordinated Mo sites are obtained or 2) c-MoS₂ goes to 0 V vs. RHE or open circuit potentials after HER, where the high thermodynamic decomposition driving force induces Mo dissolution (*ACS Energy Lett.* 2021, 6, 6, 2268–2274). At high current densities, the higher formation of undercoordinated Mo³⁺ hydride sites (see Scheme 2 main manuscript, steps I’ and II’) leads to the accelerated dissolution via stoichiometric loss of Mo and S (see step VI’ in Scheme 2). The differences in c-MoS₂ stability at low and high current densities were discussed in the manuscript, included here for ease of convenience:

*‘Indeed, c-MoS₂ and [Mo₃S₁₃]-based catalysts presented, at low HER rates, consistently different stabilities across different HER rates to those of c-MoS₂, regardless of the operating conditions (Figures S20-S22). **However, the stability crossover found when comparing low***

and high HER rates for c-MoS₂ (Mo S-number ~10⁷ vs ~10⁵, see Figure 5 and 8) suggests that intrinsic active site stabilities are not constant, and are highly dependent on the preferential HER and dissolution pathway. Such hypothesis is reinforced when analysing the post-mortem S-to-Mo XPS ratios after long-term testing, which indicate that near-surface sulphur species are favourably retained in c-MoS₂ when compared to a-MoS_{3-x} (S-to-Mo ca. 1.5 vs. 0.4, see Table S4). This leads us to believe that, for c-MoS₂, Mo-H formation cannot solely be responsible for the HER activity. At low HER rates, given the experimental evidence of low Mo dissolution and almost negligible S loss (Figures 5, 7 and S17-S19), we hypothesize that unsaturated S²⁻ sites are primarily involved as proton-accepting groups. At higher HER rates (Figure 8), given the almost stoichiometric near-surface S-to-Mo ratios found in c-MoS₂ after long-term testing (ca. 1.5, see Table S4), we propose that Mo³⁺ hydride formation (steps I' and II' in Scheme 2) would be responsible for the HER activity as well as the dissolution pathways via stoichiometric loss of Mo and S in step VI' in Scheme 2.'

In the 'Analysis of structure-activity-stability relationships' section of the manuscript, the stability trends were additionally discussed, clearly concluding that **at high current densities a-MoS_{3-x} is the preferred catalyst:**

*'Under such criterion, c-MoS₂ would seem the most suitable catalyst at low HER rates, as the S-numbers for Mo (~10⁷) and S (~10⁶) under HER potentials are beyond this threshold. Regardless, the high instability observed for Mo moieties upon voltage reversal/shut-down would prevent any intermittent PEMWE operation in a real device. In contrast, **the stability of [Mo₃S₁₃]-based catalysts markedly increased up to S-numbers ~10⁵ and ~10⁶ under higher current densities of -10 mA cm⁻² (Figure S20-22 ESI) and -100 mA cm⁻² (Figure 8), respectively. Hence, MoS_x-based PEMWEs would initially seem bound to operate under low constant current loads when using c-MoS₂, while high-current operation even under intermittent mode would be suited for [Mo₃S₁₃]-based catalysts.'***

This was also pointed out in the conclusions of the manuscript:

'The lamellar-like c-MoS₂, structurally analogous to crystalline MoS₂ presents, at lower HER rates, a higher stability (10-100 fold) at the expense of lower activity versus [Mo₃S₁₃]-based catalysts, the latter being 10-fold more stable after long-term, high-current density operation.'

'Upon mimicking intermittent operation relevant for PEMWE coupling with renewable energy inputs, we conclude that the preferential Mo loss at non-HER potentials severely compromises any MoS_x cathode lifetime unless high HER rates are employed for [Mo₃S₁₃]-based catalysts.'

We hope that the further clarifications from our long-term HER measurements are appropriate for reviewer #3 and fully clear out any apparent contradictions.

Figure S24. Compilation of total (left) and loading-normalized (right) integrated dissolution of Mo for a-MoS_{3-x} (light blue) and c-MoS_2 (dark blue) obtained after H-cell measurements consisting of 5 hours constant galvanostatic hold at -100 mA cm^{-2} (solid columns) and after 5 hours undergoing start-up/shutdown HER holds alternating from -100 mA cm^{-2} to 0 V vs. RHE (dashed columns).

HER testing	MoS _x catalyst	Loading-normalized total Mo loss (%)	Final Mo S-numbers, H-cell
Pristine -100 mA cm ⁻² , constant 5 h hold -100 mA cm ⁻² , on-off 5 h hold Ratio, constant vs. on-off	c -MoS ₂	-	-
		6.59	9.96×10 ⁴
		7.90	3.89×10 ⁴
Pristine -100 mA cm ⁻² , constant 5 h hold -100 mA cm ⁻² , on-off 5 h hold Ratio, constant vs. on-off	a -MoS _{3-x}	0.83	2.60
		-	-
		0.72	1.10 ×10 ⁶
		1.47	2.33×10 ⁵
		0.49	4.72
		5 h hold	on-off 5 h hold
c -MoS ₂ -to- a -MoS _{3-x} ratio, loading-normalized Mo loss		9.2	5.4
		11.0	6.0
a -MoS _{3-x} -to- c -MoS ₂ ratio, final Mo S-numbers, H-cell			

Table S3. Compilation of loading-normalized total dissolution and final S-numbers from Mo quantification in *a*-MoS_{3-x} and *c*-MoS₂ thin films during long-term HER testing. Values obtained after 5 hours constant galvanostatic hold at -100 mA cm⁻² and after 5 hours undergoing start-up/shutdown HER holds alternating from -100 mA cm⁻² to 0 V vs. RHE, using an H-cell configuration. Data is graphically presented in Figures 8 and S24.

Figure 8. Compilation of Mo S-numbers obtained during long-term H-cell measurements consisting of constant HER operation (5 h, -100 mA cm⁻²; left column) and start-up/shutdown HER intermittent operation alternating (5 h, -100 mA cm⁻² to 0 V vs. RHE; right column).